# Pharmacologically reversible zonation-dependent endothelial cell transcriptomic changes with neurodegenerative disease associations in the aged brain

Lei Zhao[1,2,17], Zhongqi Li [1,2,17], Joaquim S. L. Vong[2,3,17], Xinyi Chen[1,2], Hei-Ming Lai[1,2,4,5,6], Leo Y. C. Yan[1,2], Junzhe Huang[1,2], Samuel K. H. Sy[1,2,7], Xiaoyu Tian [8], Yu Huang [8], Ho Yin Edwin Chan[5,9], Hon-Cheong So[6,8], Wai-Lung Ng [10], Yamei Tang[11], Wei-Jye Lin[12,13], Vincent C. T. Mok[1,5,6,14,15✉] & Ho Ko [1,2,4,5,6,8,14,16✉]

The molecular signatures of cells in the brain have been revealed in unprecedented detail, yet the ageing-associated genome-wide expression changes that may contribute to neurovascular dysfunction in neurodegenerative diseases remain elusive. Here, we report zonation-dependent transcriptomic changes in aged mouse brain endothelial cells (ECs), which prominently implicate altered immune/cytokine signaling in ECs of all vascular segments, and functional changes impacting the blood–brain barrier (BBB) and glucose/energy metabolism especially in capillary ECs (capECs). An overrepresentation of Alzheimer disease (AD) GWAS genes is evident among the human orthologs of the differentially expressed genes of aged capECs, while comparative analysis revealed a subset of concordantly downregulated, functionally important genes in human AD brains. Treatment with exenatide, a glucagon-like peptide-1 receptor agonist, strongly reverses aged mouse brain EC transcriptomic changes and BBB leakage, with associated attenuation of microglial priming. We thus revealed transcriptomic alterations underlying brain EC ageing that are complex yet pharmacologically reversible.

[1] Division of Neurology, Department of Medicine and Therapeutics, Faculty of Medicine, The Chinese University of Hong Kong, Shatin, Hong Kong. [2] Li Ka Shing Institute of Health Sciences, Faculty of Medicine, The Chinese University of Hong Kong, Shatin, Hong Kong. [3] Department of Chemical Pathology, Faculty of Medicine, The Chinese University of Hong Kong, Shatin, Hong Kong. [4] Department of Psychiatry, Faculty of Medicine, The Chinese University of Hong Kong, Shatin, Hong Kong. [5] Gerald Choa Neuroscience Centre, Faculty of Medicine, The Chinese University of Hong Kong, Shatin, Hong Kong. [6] Margaret K. L. Cheung Research Centre for Management of Parkinsonism, Faculty of Medicine, The Chinese University of Hong Kong, Shatin, Hong Kong. [7] Department of Biomedical Engineering, Faculty of Engineering, The Chinese University of Hong Kong, Shatin, Hong Kong. [8] School of Biomedical Sciences, Faculty of Medicine, The Chinese University of Hong Kong, Shatin, Hong Kong. [9] School of Life Sciences, Faculty of Science, The Chinese University of Hong Kong, Shatin, Hong Kong. [10] School of Pharmacy, Faculty of Medicine, The Chinese University of Hong Kong, Shatin, Hong Kong. [11] Department of Neurology, Sun Yat-Sen Memorial Hospital, Sun Yat-Sen University, Guangzhou, China. [12] Guangdong Provincial Key Laboratory of Malignant Tumor Epigenetics and Gene Regulation, Medical Research Center, Sun Yat-Sen Memorial Hospital, Sun Yat-Sen University, Guangzhou, China. [13] Guangdong Provincial Key Laboratory of Brain Function and Disease, Zhongshan School of Medicine, Sun Yat-Sen University, Guangzhou, China. [14] Chow Yuk Ho Technology Center for Innovative Medicine, Faculty of Medicine, The Chinese University of Hong Kong, Shatin, Hong Kong. [15] Therese Pei Fong Chow Research Centre for Prevention of Dementia, Faculty of Medicine, The Chinese University of Hong Kong, Shatin, Hong Kong. [16] Peter Hung Pain Research Institute, Faculty of Medicine, The Chinese University of Hong Kong, Shatin, Hong Kong. [17]These authors contributed equally: Lei Zhao, Zhongqi Li, Joaquim S. L. Vong. ✉email: vctmok@cuhk.edu.hk; ho.ko@cuhk.edu.hk

Ageing is the strongest risk factor for neurodegenerative diseases. During the ageing process, complex cellular and functional changes occur in the brain. These include neurovascular dysfunction manifesting in blood–brain barrier (BBB) breakdown associated with microscopic pathological changes of vascular cells[1,2]. Forming the innermost lining of the vasculature, brain endothelial cells (ECs) closely interact with other cell types of the neurovascular unit (NVU) to regulate diverse functions including BBB integrity[3–6], neurovascular coupling[6–10], immune signaling, and brain metabolism, all of which are altered in the aged brain[1,2,11,12]. These suggest that ECs play crucial roles in the brain ageing process that are incompletely understood. With advances in single-cell genomics, especially single-cell RNA sequencing (scRNA-seq), the molecular subtypes of brain cells including ECs that segregate along the arteriovenous axis have been identified in exquisite details[13–16]. Given the heterogeneity of EC subtypes, the genome-wide expression changes in ageing brain ECs are likely highly complex and yet to be fully revealed.

Uncovering brain EC transcriptomic changes is of paramount importance for enhancing our understanding of neurovascular dysfunction in ageing, which is increasingly recognized as a major contributor to pathogenesis in multiple neurodegenerative diseases[6,17–19]. In Alzheimer disease (AD), postmortem examination revealed that vascular pathology including lacunes, microinfarcts, and other microvascular lesions coexist with classical AD pathology (i.e. amyloid plaques and neurofibrillary tangles) in the majority of patients[20]. BBB breakdown is an early and consistent feature preceding symptom onset in AD patients[17,19,21,22]. In multiple transgenic animal models of AD harboring amyloid precursor protein mutations and/or ApoE4, neurovascular deficits also precede and accelerate the development of parenchymal amyloid deposition, neurofibrillary tangles formation, neuronal dysfunction, and behavioral deficits[23]. Furthermore, altered vascular-immune cell crosstalk also occurs in ageing and AD[24–26]. This leads to endothelial activation that may contribute to the breakdown of the BBB, microglial activation, and neuroinflammation[24,25]. In numerous other neurodegenerative diseases, such as frontotemporal dementia (FTD), amyotrophic lateral sclerosis (ALS), and Parkinson's disease (PD), functional and pathological microvascular changes occur in the respective disease-affected brain regions[6,18]. While disease-specific factors differentially contribute to neuronal degeneration, ageing-associated neurovascular dysfunction represents shared pathological components among the distinct neurodegenerative diseases. It remains to be determined how ageing differentially influences different molecular subtypes of brain ECs, resulting in neurovascular deficits and neurodegenerative disease susceptibility.

Moreover, proving the reversibility of ageing-associated transcriptomic and functional changes, preferably by pharmacological means, would bear significant implications for the development of preventive or disease-modifying therapeutics. Originally developed to potentiate glucose-induced insulin release and for the treatment of diabetes mellitus, glucagon-like peptide-1 receptor (GLP-1R) agonists have been reported to have therapeutic efficacy in multiple animal models of neurodegenerative diseases[27–31], alleviating the progression of neuropathology. In human trials, GLP-1R agonists improved brain metabolism in AD and delayed motor deterioration in PD[32,33]. GLP-1R agonist treatment reduces the incidence of cardiovascular events (e.g. myocardial infarction, stroke) in diabetic patients[34], while animal tests revealed vascular protective and immunomodulatory effects in peripheral organs[35–39]. In the brain, GLP-1R is expressed by multiple cell types including microglia and subsets of neurons[28,40,41], while peripherally it has been reported to be expressed by subsets of vascular and immune cells[38,39]. It remains

to be tested whether GLP-1R agonists may act partly via reversal of neurovascular ageing, accounting for the general applicability as potential therapeutics for multiple neurodegenerative conditions.

To gain insights into the complicated gene expression changes in the aged brain vasculature, we compare single-cell transcriptomes of brain ECs from aged and young adult mouse brains, employing arteriovenous zonation markers to classify EC subtypes. We investigate the common and specific gene expression changes in different EC subtypes along the arteriovenous axis, showing that genes related to neurovascular functionality and neurodegenerative conditions exhibit arteriovenous zonation-specific differential expression in ageing. We also reveal that some expression changes in aged mouse brain ECs may generalize to human AD and aged brains, and demonstrate that the ageing-associated transcriptomic changes are amenable to pharmacological intervention by GLP-1R agonist treatment with accompanying improvement in BBB integrity.

## Results

**Profiling EC transcriptomic changes in the aged mouse brain.** To obtain ageing-associated genome-wide expression changes of neurovascular cells, we performed high-throughput scRNA-seq of brain vascular cells from young adult (2–3 months old) and aged (18–20 months old) C57BL/6J mice ($n = 5$ mice for each group) (Fig. 1a). After quality control filtering (see Methods section), we obtained a total of 43,922 single-cell transcriptomes. These included 31,555 cells from young adult and 12,367 cells from aged mouse brains (Fig. 1a), among which there were 12,357 brain ECs, 1425 smooth muscle cells (SMC), 642 pericytes (PC), 9642 astrocytes (AC), and 11,767 microglia (MG), alongside numerous other major brain cell type clusters (Fig. 1b and Supplementary Fig. 1a). The ECs had high expression of known EC-specific marker genes (e.g. *Cldn5*, *Flt1*, *Pecam1*, and *Cdh5*) and did not express marker genes specific to other cell types (e.g. *Kcnj8*, *Acta2*, *Ctss*, *Ntsr2*, *Pdgfra*, and *Cldn11*; Supplementary Fig. 1a). We employed a probabilistic method[42] (see Methods section) to assign each of the ECs from young adult ($n = 8627$ cells) and aged ($n = 3730$ cells) mouse brains to one of six molecular subtypes with distinct arteriovenous distributions based on the expression profiles of 267 arteriovenous zonation-dependent genes in ECs[13] (Fig. 1c, Supplementary Fig. 1b, and also see Supplementary Table 1 for lists of the genes). These correspond to two putative arterial subtypes (aEC1, $n = 1141$ cells and 526 cells; aEC2, $n = 998$ and 679 cells from young adult and aged groups, respectively), capillary subtype (capEC, $n = 1587$ and 697 cells), venous subtype (vEC, $n = 798$ and 412 cells), and two subtypes with mixed distributions (arterial/venous–avEC, $n = 293$ and 229 cells; and capillary-venous EC–vcapEC, $n = 3122$ and 801 cells; Fig. 1c). For both age groups, the relative expression levels of arteriovenous zonation marker genes including those of artery/arteriole (*Bmx* and *Vegfc*), capillary (*Mfsd2a* and *Tfrc*), vein (*Nr2f2* and *Slc38a5*), and artery/vein (*Vwf* and *Vcam1*) match the respective spatial designations (Fig. 1d and Supplementary Fig. 1c). Across the two age groups, the proportions of capEC subtype were similar (~20% for both groups), while we obtained more vcapEC from the young adult than aged mouse brains (39.3% and 24.0% respectively, Fig. 1e).

To assay how ageing may commonly and differentially influence brain EC subtype gene expression in different segments of the vascular network, we calculated differentially expressed genes (DEGs) expressed in natural log(fold change) (*lnFC*) in aged relative to young adult mouse group for each of the EC subtypes (Fig. 1f–i). DEGs were considered significant only if reaching statistical significance (FDR-adjusted *P*-value < 0.05)

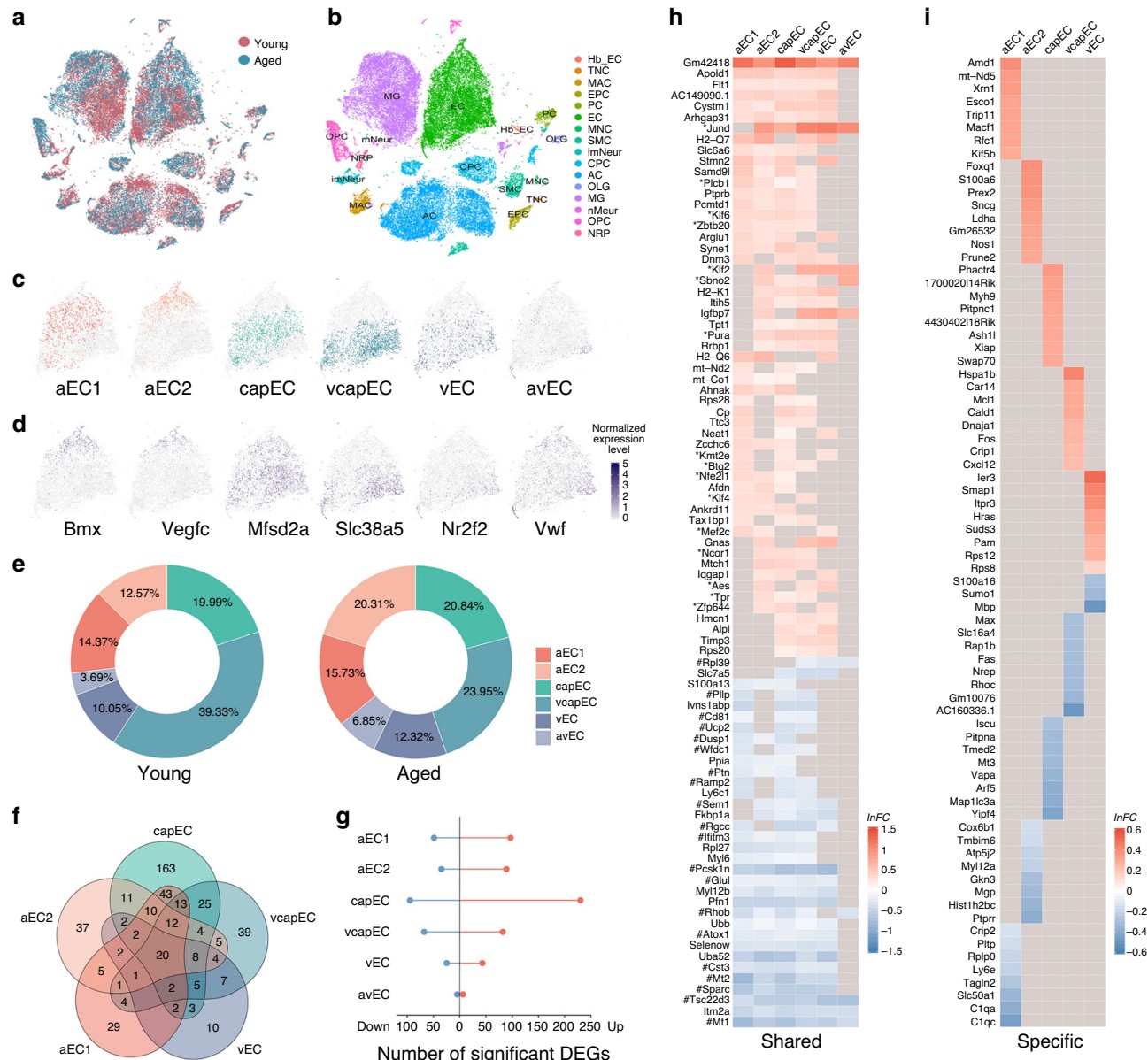

**Fig. 1 Profiling mouse brain endothelial cell transcriptomes across age. a** t-SNE visualization of single-cell transcriptomes from young adult (2–3 months old, $n = 31{,}555$ cells from five mice, brown) and aged (18–20 months old, $n = 12{,}367$ cells from five mice, blue) mouse brains. **b** Primary cell type clusters identified based on marker gene expression patterns. Cell type abbreviations: EC endothelial cell, SMC smooth muscle cell, PC pericyte, MG microglia, AC astrocyte, NRP neuronal restricted precursor, OPC oligodendrocyte precursor cell, OLG oligodendrocyte, mNeur mature neuron, imNeur immature neuron, EPC ependymocyte, CPC choroid plexus epithelial cell, Hb_EC hemoglobin-expressing vascular cell, MAC macrophage, TNC tanycyte, MNC monocyte. **c** t-SNE visualization of EC subtypes, including two arterial (aEC1, $n = 1141$ cells and 526 cells; aEC2, $n = 998$ and 679 cells from young adult and aged groups, respectively), capillary (capEC, $n = 1587$ and 697 cells), venous and capillary (vcapEC, $n = 3122$ and 801 cells), venous (vEC, $n = 798$ and 412 cells), and arterial/venous (avEC, $n = 293$ and 229 cells) subtypes. Subsequent differential expression analyses for each EC subtype presented were based on these cells (i.e. sharing the same sample sizes). For visualization, different colors are used for the subtypes, and 6000 cells were subsampled and shown for better clarity. **d** Expression patterns of arterial (*Bmx*, *Vegfc*), capillary (*Mfsd2a*), venous (*Slc38a5*, *Nr2f2*), and arterial/venous (*Vwf*) marker genes in ECs visualized by t-SNE. For clarity, 6000 cells were subsampled and shown. **e** Proportions of EC subtypes obtained from each age group. Scale bar: normalized expression level. **f** Venn diagram showing the overlap of significant differentially expressed genes (DEGs) (FDR-adjusted *P*-value < 0.05 and |*lnFC*| > 0.1) between five EC subtypes. **g** Numbers of significant upregulated (red dots) and downregulated (blue dots) DEGs for each EC subtype. **h** Heatmap of shared (significant in ≥3 EC subtypes) upregulated (upper panel) and downregulated (lower panel) DEGs in the aged brain. Remarks: asterisks denote transcription factor/regulatory genes, and hashtags denote stress response genes by the Gene Ontology annotation. **i** Heatmap of EC subtype-specific upregulated (upper panel) and downregulated (lower panel) DEGs (i.e. significant differential expression with adjusted *P*-value < 0.05 and |*lnFC*| > 0.1 in only one EC subtype). Up to eight are shown for each EC subtype.

with $|lnFC| > 0.1$. Among the EC subtypes, aged capEC had the largest number of significant DEGs while avEC had the least expression changes (capEC: 325; vcapEC: 150; aEC1: 146; aEC2: 124; vEC: 68; avEC: 11; Fig. 1f, g). While the low number of aged avEC DEGs may be due to a comparatively smaller sample size limiting power of detection, we did not find a monotonic relationship between number of cells sampled and number of significant DEGs for the other five EC subtypes (Supplementary Fig. 2a). Although we sampled more vcapEC than capEC, capEC exhibited substantially more significant differential expressions than vcapEC (Fig. 1f, g and Supplementary Fig. 2a). Indeed, vcapEC had similar DEG number as arterial ECs with only approximately halved cell numbers (Fig. 1f, g and Supplementary Fig. 2a). For all EC subtypes in the aged brain, we noted more upregulated than downregulated DEGs (Fig. 1g), in line with a recent report[43]. Altogether, the EC subtypes' significant DEGs constituted a total of 469 unique genes. In all, 22.0% (103 out of 469) of the DEGs were not significant for all EC pooled (Supplementary Fig. 2b), while those significant tend to have magnitudes of expression changes underestimated (Supplementary Fig. 2c). These highlighted the importance of fine subtype classification in addition to the localization of expression changes with respect to zonation. In total, 192 significant DEGs were shared by more than one EC subtype; almost all (191 out of 192, 99.5%) had conserved directionality of change across different EC subtypes.

A set of 55 shared genes upregulated in multiple EC subtypes (≥3 subtypes) were identified, among which only 7 were significant in at least five EC subtypes (Fig. 1h). Among the shared upregulated DEGs, there was an abundance of transcription factors or regulatory genes (e.g. *Btg2*, *Klf2*, *Ncor1*, *Sbno2*, and *Zbtb20*; Fig. 1h). A smaller set of 34 commonly downregulated genes (≥3 EC subtypes) was identified, among which 15 were shared by at least five aged brain EC subtypes (Fig. 1h). Several of the shared downregulated genes (e.g. *Cst3*, *Pcsk1n*, and *Sparc*) were previously reported to be generally downregulated in EC and other aged brain cell types[44] (Fig. 1h). Numerous stress response genes (e.g. *Atox1*, metallothionein genes *Mt1* and *Mt2*, *Sem1*, and *Tsc22d3*) were present amongst the commonly downregulated genes (Fig. 1h). A total of 278 significant DEGs were specific to only one EC subtype (195 upregulated and 83 downregulated), with 163 being capEC DEGs (120 upregulated and 43 downregulated; Fig. 1f, i) and the rest specific to one of the other EC subtypes except avEC (Fig. 1f, i). Therefore, brain ECs have both shared and divergent ageing-associated differential expressions, with the most prominent changes occurring in capillary ECs.

**Functional implications of the EC transcriptomic alterations**. Age-dependent endothelial dysfunction is a major contributor to neurovascular dysfunction. To identify altered functional pathways implicated in the aged brain EC expression changes, we next performed pathway analysis on significant DEGs of each EC subtype (except avEC due to a low number of significant DEGs; Fig. 2a). Among capEC DEGs, we found significant enrichment of genes associated with adherens junction (AJ; e.g. *Afdn*, *Ctnna1*, *Iqgap1*, *Lef1*, and *Ptprb*) and tight junction (TJ; e.g. *Arhgef2*, *Cgnl1*, *Nedd4*, *Ocln*, and *Ppp2r2a*; Fig. 2a, b). Prompted by this observation, we further examined patterns of expression changes of AJ, TJ component-encoding genes, and genes that regulate BBB transcellular transport in different brain EC subtypes. The claudin 5-encoding gene *Cldn5* has been reported to be downregulated in ECs in general[44]. In our data, a significant downregulation of *Cldn5* occurred only in aged vcapEC (Fig. 2b). We also found a downregulated expression of *Mfsd2a* in aged capEC,

which encodes the major facilitator superfamily domain-containing protein 2 (MFSD2A, or sodium-dependent lysophosphatidylcholine symporter 1) and is crucial for limiting transcytosis at the BBB[45] (Fig. 2b).

In addition to AJ- and TJ-associated genes, numerous growth factors, immune, and cell-to-cell junction signaling pathways are also important regulators of neurovascular functions and BBB integrity. TGF-β signaling-related genes were significantly enriched in the DEGs of all five aged EC subtypes analyzed (e.g. upregulated: *Klf6*, *Sik1*, *Skil*, *Smad7*, and *Tgfb2*; downregulated: *Fkbp1a* and *Nedd8*; Fig. 2a, b). The upregulation of VEGF-signaling pathway genes also impacts brain ECs of all subtypes (e.g. *Flt1*, *Kdr*, *Mef2a*, and *Timp3*; Fig. 2a, b). Furthermore, the enrichment of immune/cytokine signaling-related gene sets was observed for all vascular segments. These include genes mediating interferon gamma (IFNγ) response in aEC1 and vEC (e.g. downregulated: *Bst2*, *Ifitm3*, and *Sumo1*), toll-like receptor signaling in aEC2 and vcapEC (e.g. upregulated: *Atf3*, *Fos*, *Jun*, and *Jund*), as well as interleukin signaling in capEC, vcapEC, and vEC (e.g. upregulated: *Ccnd1*, *Egr1*, *Itgb1*, and *Mcl1*; downregulated: *Cdkn1a*). Notably, VEGF and several of the interleukins for which signaling changes were implicated have reported detrimental effects on EC barrier in vitro or BBB integrity in vivo in disease models[46–50], indicating that their appropriate signaling are crucial for BBB maintenance. Overall, capEC had the most prominent neurovascular regulatory signaling pathways enrichment, which also included Eph-ephrin/EphB-ephrinB signaling (e.g. upregulated: *Ap2b1* and *Fn1*; downregulated: *Gng5* and *Hnrnpk*) associated genes (Fig. 2a, b). Elevated *Vcam1* expression has been reported to mediate leukocyte tethering and endothelial activation by a small subset of *Vcam1*-expressing ECs in the aged hippocampus[25]. Possibly limited by the relatively small cell numbers and/or differences in regional sampling, cell isolation, and sequencing protocols, we did not find increased *Vcam1* expression in aged brain avEC (Supplementary Fig. 3a, b; note that *Vcam1* served as one of the marker genes of avEC, see Supplementary Fig. 1b, c). Instead, we found significant enrichment of VCAM1-mediated immune cell transmigration pathway among capEC-upregulated genes (e.g. *Arhgap5*, *Pak2*, *Rdx*, and *Vcl*; Fig. 2a, b).

Altered energy metabolism may also be a key contributor to brain endothelial ageing[51]. We found that respiratory electron transport chain/ATP synthesis (e.g. cytochrome oxidase subunit genes *Cox6c* and *Cox7b*, *Ucp2*, multiple ATP synthase, NADH: ubiquinone oxidoreductase genes), and glucose/energy metabolism (e.g. *Hmgcs2*, *Pea15a*, and multiple SLC transporter genes) associated genes were most enriched in the downregulated DEGs of aged capEC, and to a lesser extent in aEC1 and vcapEC (Fig. 2a, b). Of note, *Slc2a1* (encoding the glucose transporter GLUT1) was found downregulated in multiple EC subtypes (Fig. 2b). Therefore, while impaired energy metabolism has been most strongly implicated in capEC, other vascular segments are also likely impacted in the aged brain.

By RNA fluorescent in situ hybridization (FISH) in brain slices at both ages, we confirmed endothelial localization for a subset of the DEGs (including *Afdn*, *Lef1*, *Ptprb*, *Mfsd2a*, *Smad7*, *Flt1*, *Ifitm3*, *Jund*, *Slc9a3r2*, and *Nostrin*; Supplementary Fig. 4). We were however not able to quantitatively verify by RNA FISH, the relatively small magnitudes of expression changes found by scRNA-seq (Figs. 1h, i and 2b). Therefore, we next performed quantitative PCR (qPCR) for selected genes in whole-brain ECs isolated by immunopanning (Fig. 2c, also see Methods section) from the two age groups. This verified the altered expression of several genes involved in key regulatory pathways of neurovascular functions (e.g. VEGF and TGF-β signaling-related genes, key transporter genes), including both aged brain EC-upregulated

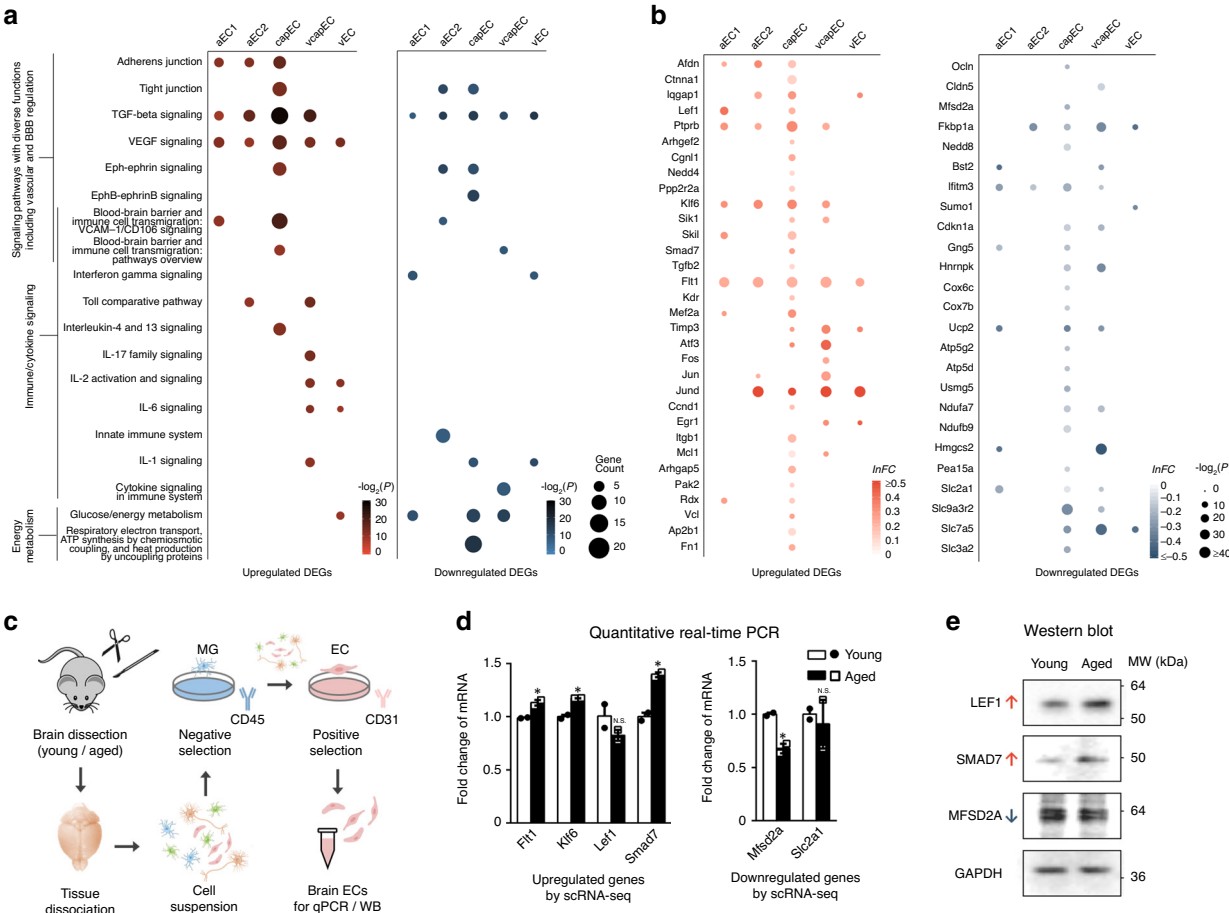

**Fig. 2 Pathway analysis and differential expression validation. a** Dot plots of important signaling pathways whose functions include vascular and BBB regulation, immune/cytokine signaling, respiratory electron transport chain, and glucose/energy metabolism pathways with significant enrichment in the aged brain for upregulated (left panel) and downregulated (right panel) EC DEGs. **b** Dot plots showing the differential expression profile of selected upregulated (left panel) and downregulated (right panel) genes in aged brain ECs associated with enriched pathways shown in **a** in the different EC subtypes. **c** Schematics of validation experiments by quantitative PCR and western blot for selected DEGs found by scRNA-seq (including *Flt1*, *Klf6*, *Lef1*, *Smad7*, *Mfsd2a*, and *Slc2a1*), in whole-brain ECs isolated by immunopanning. **d** Expression changes of selected genes in immunopanned ECs by quantitative PCR (for each gene, quantified as fold change relative to young adult group mean, error bars represent S.E.M.; *Flt1*: P = 0.036; *Klf6*: P = 0.011; *Lef1*: P = 0.28; *Smad7*: P = 0.012; *Mfsd2a*: P = 0.022; *Slc2a1*: P = 0.73; asterisks in the plot indicates P < 0.05 for easy visualization, two-sided unpaired *t*-test without adjustment for multiple comparisons; for each group, samples from *n* = 3 mice were pooled and measured in duplicate, resulting in two measurement values for each gene). **e** Western blot assays of the encoded proteins of a subset of aged brain EC-upregulated (LEF1, SMAD7) and downregulated (MFSD2A) genes in immunopanned ECs (samples pooled from three mice for each group). Source data underlying **d**, **e** are provided as a Source Data file.

(*Flt1*, *Klf6*, and *Smad7*) and downregulated (*Mfsd2a*) genes (Fig. 2d). Western blot (WB) assays on immunopanned ECs further validated the expression changes of a subset of these genes (including LEF1, SMAD7, and MFSD2A) at the protein level (Fig. 2e).

Collectively, these results revealed a complex pattern of ageing-related transcriptomic changes in specific EC subtypes that may impact neurovascular functions, especially in capEC. Important subsets of the DEGs were related to immune/cytokine signaling and energy metabolism, and may jointly contribute to age-dependent susceptibility to neurodegeneration.

**Disease associations of aged brain EC transcriptomic changes.** Genome-wide association studies (GWAS) identified many associated genetic variants of cerebrovascular and neurodegenerative diseases with a vascular component. We asked whether some of these genes would have ageing-associated expression changes in ECs (see Supplementary Table 2 for disease-associated gene lists and their sources used for this analysis). Indeed, among

the DEGs of aged brain ECs, we found a total of 40 genes associated with one or more of 7 neurovascular or neurodegenerative disorders, out of the 11 diseases examined (Fig. 3a). In total, 32 of these DEGs were specific to ECs, with no significant differential expression found in SMC, AC, or MG in our dataset (Fig. 3a). Overall, 31 disease-related genes identified were significant DEGs of capEC (Fig. 3a), and 23 are AD-associated (see Supplementary Table 3 for disease association summary). Interestingly, the abundance of AD GWAS genes among the DEGs was not merely a result of a large number of AD GWAS genes, as they were significantly overrepresented in aged capEC DEGs (Fig. 3a, number of overlapped genes = 18; P = 0.022, hypergeometric test). While numerous risk genes of other neurovascular and neurodegenerative diseases also overlapped with aged EC DEGs, they did not reach statistical significance for overrepresentation in any EC subtypes (Fig. 3a, e.g. number of overlapped genes for stroke genes and capEC DEGs = 4; P = 0.067). Notably, we found an upregulation of *Cd2ap* (Fig. 3a), which has previously been shown to be required for the maintenance of normal BBB[52] and

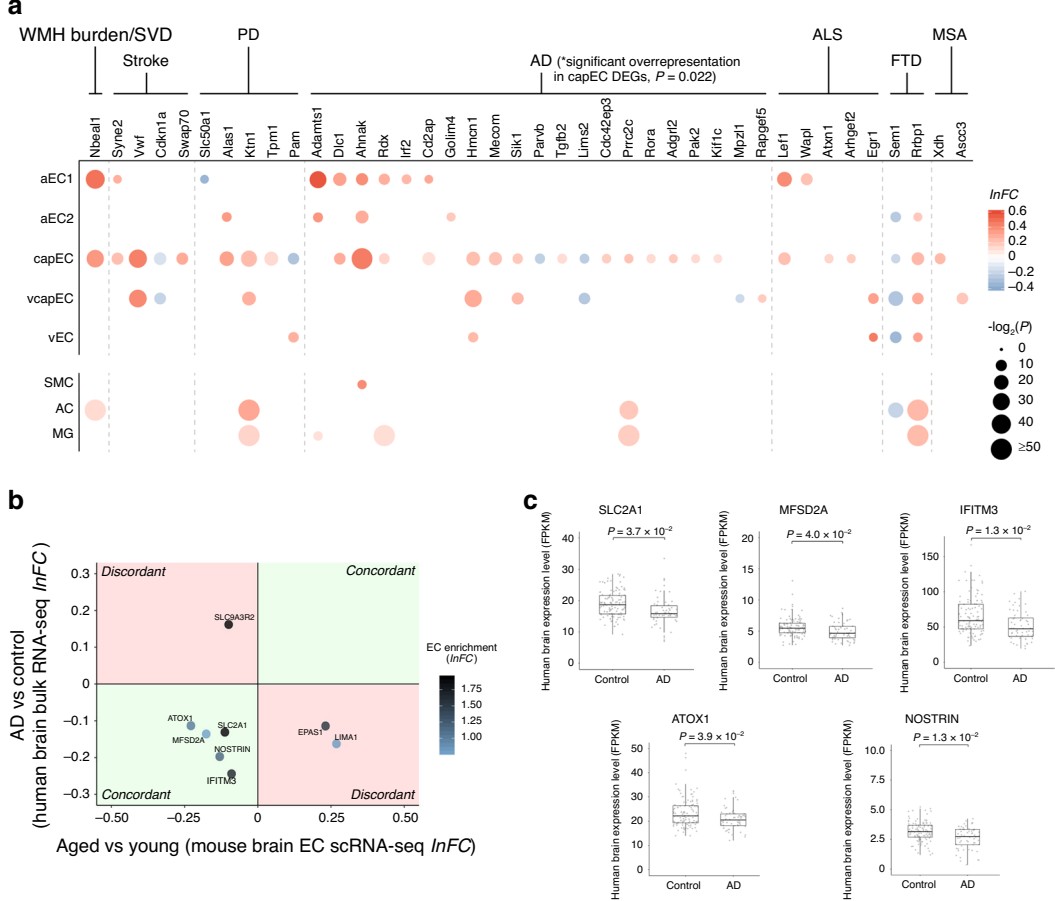

**Fig. 3 Disease associations of aged brain endothelial cell differential expressions. a** Differential expression profiles of aged brain EC subtype DEGs whose human orthologs are GWAS genes of cerebrovascular or neurodegenerative diseases examined. capEC DEGs had the most overlap with a significant overrepresentation of AD GWAS genes ($P = 0.022$, hypergeometric test). The differential expressions of these genes in smooth muscle cell (SMC), astrocyte (AC), and microglia (MG) are also shown. Disease abbreviations: WMH burden white matter hyperintensity burden, SVD cerebral small vessel disease, PD Parkinson's disease, AD Alzheimer disease, ALS amyotrophic lateral sclerosis, FTD frontotemporal dementia, MSA multiple systems atrophy. **b** Human AD brain differential expressions relative to age-matched control subjects (FDR-adjusted $P$-value < 0.05, $y$-axis) plotted against expression changes in pooled aged mouse brain ECs ($x$-axis), showing aged mouse brain EC DEGs whose human orthologs had concordant (first and third quadrants, light green) or discordant (second and fourth quadrants, light red) expression changes in human AD brains. Only DEGs with at least twofold EC-enrichment ($lnFC$ of EC expression relative to other cell types >0.7) are shown, with color of dots representing the degree of enrichment. **c** Human brain expression levels of genes with concordant expression changes in human AD and aged mouse brains ($n = 111$ samples from 30 control brains, 58 samples from 16 AD brains; two-sided unpaired $t$-test with FDR-adjustment for multiple comparisons). Black horizontal line: median; upper and lower bounds of box: 75th and 25th percentiles respectively; upper and lower bounds of vertical lines: upper quartile + 1.5 × interquartile range or maximum (whichever is smaller), and lower quartile − 1.5 × interquartile range or minimum (whichever is larger), respectively.

its loss-of-function variants may contribute to AD pathogenesis by impairing BBB integrity. We also noted the upregulation of *Dlc1* in aged capEC (Fig. 3a), which has also been identified as an upregulated gene in human AD patient entorhinal cortex ECs[53]. Our data thus support that AD has a unique vascular ageing component, especially in the capillary endothelium, among other neurodegenerative diseases.

We then attempted to identify aged mouse brain EC subtype DEGs with the same expression changes in human AD brains. We calculated DEGs for human AD versus control brains with no dementia (aged 78 – over 100 years old), from the Allen Brain Institute Aging, Dementia and Traumatic Brain Injury study dataset of human brain bulk RNA sequencing[54,55]. After selecting for EC-enriched genes, the directionality of change of the human AD DEGs was compared to the EC subtype DEGs. This revealed five EC-enriched genes with the same changes found in aged mouse and human AD patient brains (Fig. 3b, c). Manual review of the Human Protein Atlas[56], Allen Brain Atlas human brain

single-nucleus RNA-seq data[57] and available literature confirmed human brain EC expression for four of these genes (Supplementary Table 4). Interestingly, all five genes had concordant downregulation in aged mouse and AD brains (Fig. 3b, c), whereby several of them have important functional roles at the BBB. For instance, consistent with previous reports, reduced expression of *Slc2a1* in AD brains was found[58,59] (Fig. 3b, c). On the other hand, the aged mouse brain capEC-downregulated DEG *Mfsd2a* (Fig. 2b) also had reduced expression in AD brains compared to non-demented aged human brains (Fig. 3b, c), suggesting a possible contribution to BBB breakdown in AD. *Ifitm3*, an IFN-responsive gene that plays an important role in restricting viral invasion[60] was also found downregulated in multiple aged mouse brain EC subtypes (Fig. 2b) and human AD brains (Fig. 3b, c). We also carried out a similar comparative analysis using the Mayo Clinic AD brain bulk RNA-seq dataset[61]. This identified the concordant upregulation of four EC-enriched genes, including *Dlc1*, among others (Supplementary Fig. 5a, b).

The difference of genes found with the Allen Brain Institute and Mayo Clinic datasets may reflect the heterogeneity of human AD brain datasets due to differences in patient cohorts, brain region sampling, specific tissue processing, and sequencing protocols adopted.

Therefore, in addition to the overrepresentation of AD GWAS genes, we have found a multitude of similar expression changes in the normal aged mouse and human AD brains that are functionally important and may contribute to degenerative changes. In the human AD brain, a decrease in neuronal population may lead to exaggerated upregulation while masking downregulation of non-neuronal genes detected by bulk RNA sequencing. The expression of downregulated genes we found may therefore be more prominently reduced in the human AD brain ECs, while the upregulated genes are yet to be further validated.

**Comparing EC expression changes in aged human brains.** We speculated that some of the transcriptomic changes found in aged mouse brain ECs are conserved and detectable in normal human aged brains with bulk RNA sequencing data, available from the Genotype-Tissue Expression (GTEx) project database[62]. We performed two types of comparisons, by dividing the data in the database into two age groups (≥60 versus <60 years old) or regression of expression level with respect to age (range: 46–70 years old). This revealed a total of 41 human orthologs of aged mouse brain EC-enriched DEGs that exhibited statistically significant differential expression across age in the human brain (Supplementary Fig. 6a, b). Of note, the majority of these DEGs were upregulated in aged human brains (90.2%, 37 out of 41; Supplementary Fig. 6a, b), and most of these genes have confirmed human brain EC expression (95.1%, 39 out of 41, see Supplementary Table 4). In all, 20 human homologs of upregulated aged mouse brain EC DEGs had a concordant increase in expression in aged human brains (Supplementary Fig. 6a, b). These included numerous BBB regulation-associated genes (e.g. *Iqgap1*, *Ptprb*, *Timp3*, and *Flt1*) and AD-associated genes (including the P-glycoprotein gene *Abcb1a*, *Adamts1*, *Ahnak*, and *Mecom*; Supplementary Fig. 6a–d). However, 19 EC-enriched genes had different directionality of change in human aged brains, among which most (89.5%, 17 out of 19) had increased expression in aged human brains but downregulated in aged mouse brain ECs (Supplementary Fig. 6a–d). These included several genes with important functions at the BBB, such as *Cldn5*, *Slc2a1*, and *Ifitm3* (Supplementary Fig. 6a–d). Additionally, *Mfsd2a* exhibited a trend of upregulation in aged human brains (Supplementary Fig. 6c). We only noted one gene, *Epas1*, that exhibited concordant upregulation in aged mouse and human brains but downregulated in the human AD brain. However, a previous study reported its upregulation in the human AD entorhinal cortex ECs[53]. Further studies are required to explore and identify more such genes, which may represent important divergent changes in AD from normal ageing. Like AD brains, expression quantification of the ageing human brain by bulk RNA sequencing may be confounded by a decrease in neuronal population, exaggerating upregulations and masking downregulations of non-neuronal genes. This may partly account for the general trend of upregulation and some of the discordances found.

**GLP-1R agonist treatment potently reverses brain EC ageing.** We next asked if the complex subtype-dependent transcriptomic alterations of aged brain ECs are amenable to pharmacological intervention with improvements in neurovascular deficits. Since transcriptomic changes involving neurovascular regulatory,

immune/cytokine signaling and energy metabolism pathways may all impact EC function and the BBB, we went on to assay ageing-associated BBB leakage and tested its reversibility. GLP-1R agonists had reported therapeutic efficacies in both animal models and human clinical trials in neurodegenerative conditions[27–33]. At other body sites (e.g. the myocardium), GLP-1R agonist treatment has been reported to improve endothelial function, energy metabolism, and exert immunomodulatory effects[35–39,63,64]. We thus postulated that GLP-1R agonists might have neurovascular protective effects that partly explains its general applicability as potential therapeutics for multiple neurodegenerative conditions with ageing-associated neurovascular defects as a shared pathological component.

To assay BBB integrity, we performed in vivo two-photon imaging of extravasation of intravenously injected fluorophore-conjugated dextran in the cerebral cortex. Consistent with a previous report[25], we did not observe extravasation of large molecular weight (70 kDa) dextran in either age groups (Fig. 4a). Compared to young adult (2–3 months old) mice, aged (18–20 months old) mice showed 15.8-fold increase of 40 kDa TRITC-conjugated dextran (TRITC-dextran) extravasation in the cerebral cortex (see Methods section), signifying increased BBB leakage (Fig. 4a, b; volume of extravasated TRITC-dextran in aged vs young adult group, $P = 2.2 \times 10^{-5}$, three image stacks were acquired to obtain the mean for each animal, $n = 3$ mice for each group, one-way ANOVA with Tukey's post-hoc test). Consistent with our hypothesis, 4–5-week treatment of aged mice by the GLP-1R agonist exenatide (5 nmol/kg/day, intraperitoneal injection starting at 17–18 months old) reduced TRITC-dextran extravasations in the cerebral cortex by more than 50% (Fig. 4a, b, $P = 5.9 \times 10^{-4}$ compared to 18 m.o. untreated group, 3 image stacks were acquired to obtain the mean for each animal, $n = 3$ mice for each group, one-way ANOVA with Tukey's post-hoc test; also see Supplementary Fig. 7 for an independent batch of experiments verifying the therapeutic effect of exenatide over saline vehicle). We did not find differences in vascular length density across the different groups (Fig. 4c, average cortical vascular length density ± S.E.M. = 0.95 ± 0.11, 0.94 ± 0.03, 0.84 ± 0.11 × 10^7 μm per mm^3 in young adult, aged and exenatide-treated aged groups respectively, $P = 0.68$, one-way ANOVA), indicating that the BBB leakage comparisons were not confounded by a potential age-related change in vascular density[65].

Remarkably, the improvement of BBB integrity was associated with reversal of ageing-associated gene expression changes in brain ECs (Fig. 4d, e and Supplementary Fig. 8). For statistically significant capEC DEGs (i.e. adjusted $P$-value < 0.05 regardless of $|lnFC|$), exenatide treatment reversed the expression changes of 76.4% upregulated (217 out of 284, $P = 3.0 \times 10^{-11}$ compared to chance level) and 63.4% downregulated DEGs (85 out of 134, $P = 6.9 \times 10^{-7}$ compared to chance level; Fig. 4d, left upper panel, $r^2 = 0.27$; $P = 7.2 \times 10^{-30}$). This reversal effect also applies to other EC subtypes (Supplementary Fig. 8). Classifying the DEGs by functionality, strong reversal was found for neurovascular regulatory (Fig. 4d, right upper panel, 76.7% of DEGs reversed, $r^2 = 0.59$; $P = 8.51 \times 10^{-13}$) and immune/cytokine signaling (Fig. 4d, left lower panel, 66.7% of DEGs reversed, $r^2 = 0.41$; $P = 7.99 \times 10^{-4}$) related genes identified from pathway analysis in capEC. The majority of downregulated respiratory electron transport chain- and glucose/energy metabolism-associated genes were upregulated by exenatide treatment in capEC (Fig. 4d, right lower panel, 80% upregulated, $P = 0.02$ compared to chance level). Finally, the observation also generalized to the subcategory of DEGs whose human orthologs are AD-associated (Fig. 4e, upper panel, 68.4% of DEGs reversed, $r^2 = 0.08$; $P = 0.24$) in capEC, or with concordant changes in the normal aged or AD human brains (Fig. 4e, lower panel, 75.9% of DEGs reversed,

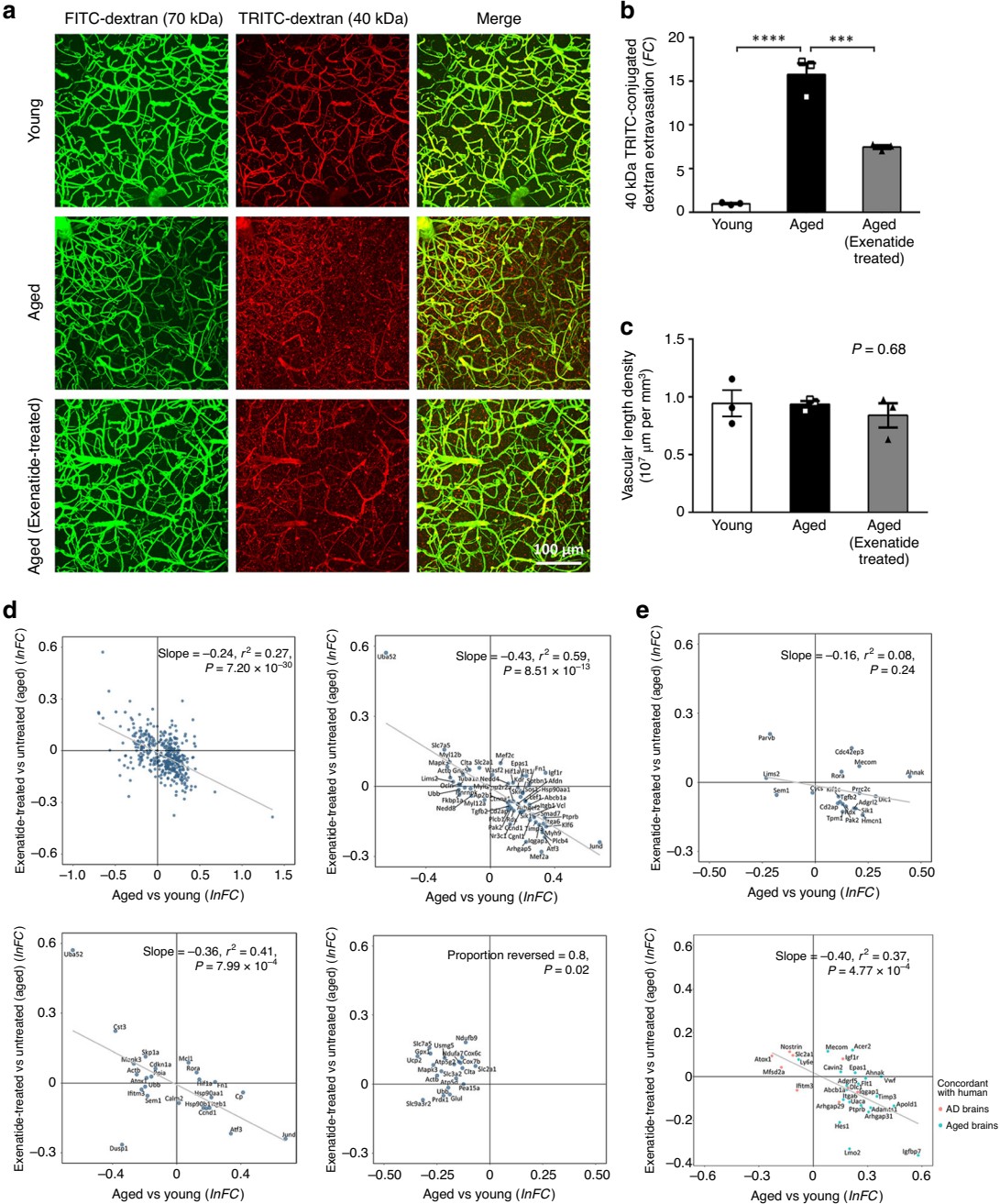

**Fig. 4 Functional and transcriptomic reversal of ageing-associated endothelial changes by exenatide treatment. a** Three-dimensional rendered images (top view) of in vivo two-photon imaging of cerebral vasculature and blood–brain barrier (BBB) leakage in the mouse somatosensory cortex by co-injection of 70 kDa FITC-conjugated dextran (FITC-dextran, green) and 40 kDa TRITC-conjugated dextran (TRITC-dextran, red). FITC-dextran remained in the vasculature and allowed reconstruction of vessels, while extravasation of TRITC-dextran served as an indicator of BBB leakage which was quantified for young adult, aged and exenatide-treated aged mouse groups. **b** Volumetric quantification of TRITC-dextran extravasation showing BBB breakdown in aged (18–20 months old) relative to young adult mice (2–3 months old) (mean fold change (*FC*) in volume of extravasated TRITC-dextran relative to young adult group ± S.E.M. = 15.8 ± 1.3; $P = 2.2 \times 10^{-5}$ for aged vs young adult mouse group, 3 image stacks were acquired to obtain the mean for each animal, $n = 3$ mice for each group, one-way ANOVA with Tukey's post-hoc test), which was significantly reduced by exenatide treatment (5 nmol/kg/day I.P. for 4–5 weeks starting at 17–18 months old, mean fold change relative to young adult group ± S.E.M. = 7.5 ± 0.2; $P = 5.9 \times 10^{-4}$ for exenatide-treated vs untreated aged mouse group, 3 image stacks were acquired to obtain the mean for each animal, $n = 3$ mice for each group, one-way ANOVA with Tukey's post-hoc test). **c**, Cortical vascular length density from the three experimental groups (mean cortical vascular length density ± S.E.M. = 0.95 ± 0.11, 0.94 ± 0.03, 0.84 ± 0.11 × $10^7$ μm per $mm^3$ in young adult, aged and exenatide-treated aged groups respectively, $P = 0.68$, one-way ANOVA). Source data underlying **b**, **c** are provided as a Source Data file. **d** Reversal of brain capillary EC overall (left upper panel), vascular regulatory (right upper panel), immune/cytokine signaling (left lower panel) and energy metabolism (right lower panel) associated gene expression changes by exenatide treatment in aged mouse. **e** Reversal of aged mouse brain EC differential expressions whose human orthologs are AD GWAS genes in capEC (upper panel), or had concordant changes in human normal aged or AD brains in all ECs pooled (lower panel).

$r^2 = 0.37;\ P = 4.77 \times 10^{-4}$) in all ECs pooled. We have thus shown that despite the complexity, the transcriptomic and functional changes in aged brain ECs are amenable to pharmacological interventions.

**Associated attenuation of age-related microglial priming.** BBB leakage could lead to microglial activation[66], while activated microglia could reciprocally mediate BBB sealing in acute cerebrovascular injury[67,68], as well as BBB breakdown via cytokine production and astrocyte endfeet phagocytosis in ageing or chronic pro-neuroinflammatory conditions[67,69]. Numerous ageing- and neurodegenerative disease-enriched microglial genes, which may play a role in priming microglia to be activated, have been previously reported[70,71]. We thus examined the expression changes of these and activation-associated genes in the aged brain microglia, and after GLP-1R agonist treatment. Immunostaining revealed a trend of higher IBA1+ microglia density in ageing, in line with previous reports[25,72], which was reduced with exenatide treatment (Supplementary Fig. 9a, b, mean IBA1+ cell density ± S.D. = $141 \pm 36$ per mm$^2$ in young adult group, $171 \pm 34$ per mm$^2$ in aged group, and $154 \pm 28$ per mm$^2$ in exenatide-treated aged group brain sections; $n = 9$, 8, and 12 imaged hippocampal and cortical regions of 4, 3, and 3 mice from young adult, aged, and exenatide-treated aged groups respectively; $P = 0.17$, one-way ANOVA). Interestingly, scRNA-seq revealed that the upregulation of various ageing- and neurodegenerative disease-enriched genes in microglia[70,71] was attenuated by exenatide treatment (Supplementary Fig. 9c, d). These included the classical AD risk gene *Apoe*, among others (*Ccl4, Ccl6, Cd9, Cd52,* and *Timp2*; Supplementary Fig. 9c, d). Therefore, the reversal of ageing-associated EC transcriptomic changes and BBB leakage by GLP-1R agonist treatment, was also associated with an attenuation of microglial ageing- and disease-associated signature.

## Discussion

Advances in single-cell transcriptomic profiling have led to the introduction of the vascular zonation concept, which enables the fine classification of EC subtypes along the arteriovenous axis based on gene expression differences, highlighting their potential functional distinctions[13]. This opens up the possibility of studying how distinct EC subtypes may be selectively vulnerable in neurovascular and neurodegenerative diseases[6]. In this study, we focus on uncovering transcriptomic changes in aged brain ECs subtypes that may underlie neurovascular dysfunction and increased susceptibility to ageing-associated neurodegenerative conditions. This could potentially advance our understanding of normal ageing changes as a common factor in these diseases, paving the way for further studies of how disease-specific factors lead to differentially compromised NVU function in different conditions.

By classifying EC transcriptomes into six major subtypes according to the reported expression patterns of arteriovenous-variable genes, we uncovered both common and specific ageing-associated transcriptomic changes in the different EC subtypes. Among BBB-associated pathways with enrichment of expression changes, altered TGF-β, VEGF, and immune/cytokine signaling appear to impact all vascular segments, while others exhibit subtype-specific changes that notably all impact capEC[73]. Despite the partially overlapping zonation of capEC and vcapEC in capillaries, ageing-related expression changes are more prominent in capEC. On the other hand, although energy metabolism-related pathway genes had altered expression in both arterial and capillary ECs, joint-downregulation of multiple cytochrome oxidase subunits, ATP synthase, NADH:ubiquinone oxidoreductase, and SLC transporter genes only occur in capillary ECs. Our findings thus revealed that the fine zonation classification of ECs is directly relevant to their gene expression and functional changes in the aged brain.

Importantly, we found that VEGF and various cytokine-related signaling pathways are altered in aged capEC and vcapEC. These pathways have been implicated in BBB disruption in ageing, multiple sclerosis and related disease models, or direct injection of the corresponding cytokines[46–50]. Apart from ageing, enrichment of immune/cytokine signaling-associated genes at the BBB has also been reported in several neurological disease models[74]. Recently, alterations of vascular-immune cell interactions and glial cells in the aged brain are becoming better understood. These include VCAM1-mediated leukocyte tethering and endothelial activation in the hippocampus[25,43], and infiltrated T-cell-mediated IFNγ pathway activation in multiple cell types in the subventricular zone[24]. Suppression of EC-orchestrated immune responses has also been reported[75]. Indeed, one may speculate that in the aged brain, physiological immune responses of brain ECs may be hampered as maladaptive immune responses prevail. While age-related microglial activation is relatively well reported and studied[69,70], reactive astrocyte phenotypes with upregulation of complement and MHC-related genes, downregulation of cholesterol metabolism, as well as altered expression of synaptic regulatory genes in the aged brain are also becoming more thoroughly characterized[76,77]. Transcriptomic and associated functional changes in capEC and vcapEC we identified, therefore, can jointly contribute with these changes in the aged brain to mediate BBB breakdown, neuroinflammation and neurodegenerative disease susceptibility.

In AD, neurovascular dysfunctions manifesting in reduced cerebral blood flow, impaired neurovascular coupling, and BBB breakdown precede neurodegeneration and atrophic changes[17,18]. Interestingly, we found an overrepresentation of AD GWAS genes among the human orthologs of the aged capillary ECs DEGs. Some of these members have known functional roles in the brain vasculature (e.g. *Cd2ap*[52]) and hence more direct interpretations regarding their differential expressions. The mechanistic linkage for the majority of other identified GWAS genes to neurodegeneration remains unclear, and some differential expressions may represent adaptive changes. We noted our list did not overlap with a previous report on overrepresented AD GWAS genes among aged human brain microglia[78]. It seems plausible that the aged microglia-enriched AD GWAS genes have an influence on AD development via altering microglial responses, while the expression changes of at least a subset of the genes we found in ECs have pathogenic and/or adaptive roles in the aged brain vasculature yet to be uncovered.

Additionally, for a subset of aged mouse brain EC DEGs with important functional implications, we found similar changes in human AD brains compared to age-matched controls. Intriguingly, these mostly had downregulation of expression. The reduced expression of GLUT1 glucose transporter gene *Slc2a1* in aged mouse brains resembles that found in human AD[58,59]. GLUT1 heterozygous deletion in ECs has been shown to accelerate the development of pathology in the APP$^{\mathrm{Sw/0}}$ mouse model of AD[79]. We also found additional potential linkage of expression changes in the aged brain and AD, such as the endothelial downregulation of *Mfsd2a*. MFSD2A serves as the major transporter for brain uptake of the essential omega-3-fatty acid docosahexaenoic acid across the BBB[80]. It is essential for BBB formation and maintenance, as mice with MFSD2A knockout develop leaky BBB despite grossly normal vasculature morphology[81]. Of note, MFSD2A lipid transport function is required for the inhibition of caveolae-mediated transcytosis across the endothelium[45]. The downregulation of MFSD2A may contribute to BBB breakdown and AD pathogenesis alongside other

expression changes. These findings highlight the polygenic nature of AD, with ageing known to be the most important risk factor. Currently, AD pathogenesis studies and drug development rely heavily on mice harbouring familial AD mutant genes as experimental models and concerns often surround the generalizability of conclusions. Indeed, these AD mice are unable to model all aspects of human AD pathology. Our finding that several key loss-of-function changes at the vasculature occur both in aged mouse and human AD brains provides the scientific ground of using aged mice as an alternative model, with additional disease-mimicking factors depending on which research questions to address. For example, temporally controlled expression of mutant amyloid precursor protein and/or presenilin in aged mice may be a more accurate model for AD therapeutics testing. The mechanism(s) for the partial resemblance of aged mouse brain vascular changes to that in the human AD brain and some discordances with the normal aged human brain are yet to be determined. These could be reflecting inter-species differences, whereby the aged mouse brain vasculature is more prone to expression and functional alterations that predispose the nervous system to degenerative changes.

Ageing has long thought to be an irreversible process involving gene expression changes that impact diverse cellular processes. Strikingly, despite the complex subtype-dependent transcriptomic changes, we found a strong reversal effect of GLP-1R agonist treatment in all major EC subtypes that accompanies vascular functional improvements and attenuated age-related upregulation of disease-associated genes in microglia. This observation holds for DEGs of aged brain ECs associated with identified enriched functional pathways and AD. Although it is hard to dissociate the exact cellular pathways involved, our data suggest that correcting the expressions of immune/cytokine signaling, energy metabolism, and key neurovascular regulatory genes may all play a role. Importantly, the finding also holds for the subsets of genes with concordant changes in human AD brains (e.g. *Slc2a1* and *Mfsd2a*). GLP-1R agonists have previously been found efficacious in several animal models of neurodegenerative diseases, including AD and PD, in protecting against neuronal death or degeneration[27–31]. Evidence also suggests that these findings may be generalizable to human patients. In a phase II PD trial, weekly subcutaneous injection of exenatide halted the deterioration of off-medication motor scores over a treatment period of 48 weeks, an effect that persisted even after a 12-week washout period[33]. Another phase II study in AD showed that 6-month liraglutide treatment prevented the decline of brain glucose metabolism[32]. Given that ageing-associated NVU dysfunction represents a common component for AD and other neurodegenerative conditions, our study shows that reversal of ageing-associated changes in brain ECs and BBB protection is an important underlying mechanism that explains GLP-1R agonists' general applicability as potential disease-modifying therapeutics.

The practical pharmacological approach we demonstrated complements the recent interesting discovery that young plasma infusion can partially reverse ageing-associated hippocampal EC transcriptomic alterations[43]. Apart from the reversal of expression changes in the aged brain endothelium, GLP-1R agonist treatment and young plasma infusion appear to share several functional benefits in animal models, such as rescuing synaptic plasticity deficits and cognitive benefits in aged or AD mouse[30,82]. Further studies are required to clarify the exact loci of actions for GLP-1R agonists. As GLP-1R is broadly expressed in diverse tissues and cell types[28,40,41,56], and GLP-1R agonists can cross the BBB[83,84], the observed neurovascular benefits may depend on diverse mechanisms, such as transcriptomic reversal in aged brain ECs, reduced microglial activation, modulation of peripheral

immune cells, and altered compositions in the circulation. Indeed, our finding that GLP-1R agonist treatment reduces the expression of ageing- and neurodegenerative disease-associated transcripts in microglia, which has been reported to express GLP-1R[28], suggests that modulation of microglia-dependent neuroinflammatory pathways is an important effect of GLP-1R agonists. Certainly, both neurovascular and neuroimmune mechanisms could be underlying the efficacies of GLP-1R agonists reported in human AD and PD trials[32,33].

GLP-1R agonist treatment also leads to functional recovery in the aged BBB. This is important as BBB breakdown is an early event in the vascular two-hit hypothesis of AD pathogenesis, increasing brain tissue exposure to neurotoxins that promotes neuroinflammation, which exacerbates amyloid beta and tauopathy-associated neurotoxicity to cause synaptic and neuronal degeneration[6,17–19,85]. Since ageing is the greatest risk factor of AD and BBB breakdown in ageing can contribute to the development of neuropathology[1,2,21], we propose that in the early or presymptomatic phase of neurodegeneration, BBB protection represents a promising therapeutic strategy for preventing or delaying neurodegeneration in ageing. Further trials testing GLP-1R agonists are underway and may complement other approaches to the development of therapeutics for AD[86]. It is worth noting that however, that the brain must have adequate exposure to the anti-amyloid antibodies or other therapeutics to achieve potential therapeutic efficacy. It remains to be determined whether simultaneously administration of another pharmacological agent that improves BBB integrity in such context may be beneficial by reducing brain tissue exposure to neurotoxins and proinflammatory substances, or would hamper the therapeutic antibodies from exerting the desirable effects.

We wish to emphasize that the interpretation of transcriptomic data is often associative and not causal, and hence it is important to avoid false-positive discoveries or overinterpretation of results. Despite the shallow sequencing depth and limited by sample size, by adopting relatively stringent criteria for defining significant DEGs, we sought to identify the most prominent functional pathways impacted and their associations with neurodegenerative diseases. To establish a causal relationship between the expression changes found and altered neurovascular functions, further mechanistic studies involving enhanced expression, knockdown or knockin/out of specific genes(s) are required. It is also important to note that, despite the lack of significant overrepresentation, other neurovascular or neurodegenerative disease-associated genes with differential expression in the aged brain endothelium may still carry functional implications that are yet to be revealed (e.g. *Nbeal1*, associated with cerebral small vessel disease/stroke and most upregulated in aged arterial ECs, see Fig. 3a and Supplementary Table 3). Moreover, in our study, whole-brain vasculature isolation was performed, which could mask the regional heterogeneity of cellular subtypes and differential expressions in ageing. For diseases with regional vulnerability (e.g. ALS, FTD, and PD), targeted dissection of disease-affected regions for transcriptomic profiling of neurovascular and glial cells will be needed to uncover additional disease-associated expression changes. As we only used male mice for experimentation, future studies in animals and human subjects will also be required to examine the sex differences in ageing-associated transcriptomic changes. Indeed, it is known that aged female and male have different neurovascular and neurodegenerative disease risk profiles in ageing[87,88], while transcriptomic alterations also exhibit sex-specificity in AD brain tissues[89]. Nonetheless, we anticipate that similar analyses and approach will be applicable to reveal ageing-associated cell type-specific expression changes in the brain and other organ systems[90], and provide further insights into disease vulnerability.

## Methods

**Animal subjects.** All experimental procedures were approved in advance by the Animal Research Ethical Committee of the Chinese University of Hong Kong (CUHK) and were carried out in accordance with the Guide for the Care and Use of Laboratory Animals. C57BL/6 J mice were provided by the Laboratory Animal Service Center of CUHK and maintained at controlled temperature (22–23 °C) with an alternating 12 h light/dark cycle with free access to standard mouse diet and water. The ambient humidity was maintained at <70% relative humidity. Male mice of two age groups (2–3 months old and 18–20 months old) were used for experiments. For the treatment groups, exenatide (5 nmol/kg bw, Byetta, Astra-Zeneca LP) or saline vehicle (0.9% w/v sodium chloride) was intraperitoneally (I.P.) administered (volume: 250 μl per 30 g bw) daily starting at 17–18 months old for 4–5 weeks prior to experimentation. Separate animals were used for in vivo imaging and single-cell RNA sequencing experiments.

**Brain tissue dissociation and single-cell isolation.** We adapted a dissociation protocol optimized for single brain vascular cells isolation from both young and old mouse brains[91]. Briefly, mice were deeply anaesthetized and perfused transcardially with 20 ml of ice-cold phosphate buffered saline (PBS). Mice were then rapidly decapitated, and whole brains (except cerebellum) were immersed in ice-cold Dulbecco's modified Eagle's medium (DMEM, Thermo Fisher Scientific, US). The brain tissues were cut into small pieces and dissociated into single cells using a modified version of the Neural Tissue Dissociation Kit (P) (130-092-628, Miltenyi Biotec, US). Myelin debris was removed using the Myelin Removal kit II (130-096-733, Miltenyi Biotec, US) according to the manufacturer's manual. Cell clumps were removed by serial filtration through pre-wetted 70-μm (#352350, Falcon, US) and 40-μm (#352340, Falcon, US) nylon cell strainers. Centrifugation was performed at $300 \times g$ for 5 min at 4 °C. The final cell pellets were resuspended in 500–1000 μl FACS buffer (DMEM without phenol red (Thermo Fisher Scientific, US), supplemented with 2% fetal bovine serum (Thermo Fisher Scientific, US)).

**Single-cell library preparation, sequencing and alignment.** Single-cell RNA-seq libraries were generated using the Chromium Single Cell 3′ Reagent Kit v2 (10X Genomics, US). Briefly, single-cell suspension at a density of 500–1000 cells/μL in FACS buffer was added to real-time polymerase chain reaction (RT-PCR) master mix aiming for sampling of 5000–8000 cells, and then loaded together with Single Cell 3′ gel beads and partitioning oil into a Single Cell 3′ Chip according to the manufacturer's instructions. RNA transcripts from single cells were uniquely barcoded and reverse-transcribed within droplets. cDNA molecules were pre-amplified and pooled, followed by library construction according to the manufacturer's instructions. All libraries were quantified by Qubit and RT-PCR on a LightCycler 96 System (Roche Life Science, Germany). The size profiles of the pre-amplified cDNA and sequencing libraries were examined by the Agilent High Sensitivity D5000 and High Sensitivity D1000 ScreenTape Systems (Agilent, US), respectively. All single-cell libraries were sequenced with a customized paired-end with single indexing (26/8/98-bp for v2 libraries) format according to the recommendation by 10X Genomics. All single-cell libraries were sequenced on a HiSeq 1500 system (Illumina, US) or a NextSeq 500 system (Illumina, US) using the HiSeq Rapid SBS v2 Kit (Illumina, US) or the NextSeq 500 High Output v2 Kit (Illumina, US), respectively. An average of 21017.2 mean reads per cell (ranging: 12,665–51,665) was obtained, which detected an average of 801.4 genes per cell (range: 540–1274). The library sequencing saturation was on average 77.89%. The data were aligned in Cell Ranger (v3.0.0, 10X Genomics, US).

**Data quality control and batch effect correction.** Data processing and visualization were performed using the Seurat package (v3.0.1)[92] and custom scripts in R (v3.5.1). The raw count matrix was generated by default parameters (with the mm10 reference genome). There were 63,300 cells in the primary count matrix (see [http://cells.ucsc.edu/?ds=aging-brain], where the raw and processed data is hosted and can be visualized, also available at [https://singlecell.broadinstitute.org/single_cell/study/SCP829/aging-mouse-brain-kolab]). Genes expressed by fewer than three cells were removed, leaving 19,746 genes in total. Among these genes, 4000 high-variance genes were identified by the Seurat *FindVariableFeatures* function. To remove batch effects between different sequencing batches, we used a canonical correlation analysis (CCA)-based integration function in Seurat. The high-variance genes were used to find anchor genes as a stable reference to eliminate batch effect from the first 30 dimensions from CCA (see the same websites as above where the batch-effect corrected data is also hosted and can be visualized). After batch effect removal, the reassembled dataset was filtered to exclude low-quality cells by the following criteria: (1) <5% or >95% UMI count or gene count, or (2) proportion of mitochondrial genes >20%. For dimensionality reduction, principal component analysis (PCA) was applied to compute the first 60 top principal components. Clustering was carried out by the Seurat functions *Find-Neighbors* and *FindClusters*. The *FindNeighbors* function constructed a shared nearest neighbor (SNN) graph based on the first 50 principal components. Modularity optimization was then performed on the SNN results for clustering (resolution parameter: 1.6). We employed t-distributed stochastic neighbor embedding (t-SNE) to visualize the clustering results and included 52 initial clusters for further

analysis, among which 3 clusters with abnormally high mitochondrial gene percentages were removed.

**Cell type and subtype identification.** To identify primary cell types, we employed known cell type-specific marker genes and examined their expression levels among all 49 clusters included. We further excluded clusters with a dual-high expression of two or more cell type-specific marker genes. These included a cluster with high expression of both endothelial cell and pericyte markers, due to contamination of pericytes by endothelial cell fragments also reported in the previous literature[13,93,94]. The remaining clusters were classified into 16 primary cell types (see Fig. 1 and Supplementary Fig. 1). Numbers of ECs from each batch were: young adult group, 2220, 2843, and 3564; aged group: 324, 1827, and 1579. To identify EC subtypes, we employed a probabilistic method, *CellAssign*[42], to classify each EC into one of the six principal subtypes using 267 variable genes along the arteriovenous axis from previously published literature as prior knowledge[13]. Each cell was assigned to the subtype with the highest likelihood (see Supplementary Table 1 for full list of the 267 genes), and the algorithm was run 3 times independently. Only endothelial cells with identical subtype assignments for all three runs were included for subtype-based differential expression analysis (11,283 out of 12,357, 91.3%).

**Differentially expressed gene calculation.** After batch effect removal, quality control and cell type/subtype classification, gene count normalization and high-variance gene identification were applied to the raw data of 43,922 cells retained for further analysis. The Seurat *FindMarkers* function and the MAST package (v1.8.2) were employed for calculation of differentially expressed genes (DEG) with associated raw P-value, false discovery rate (FDR)-adjusted P-value and magnitude of change expressed in natural log of fold change (*lnFC*) for each cell type/subtype. We define significant DEGs as those fulfilling FDR-adjusted P-value < 0.05 and an absolute value of *lnFC* exceeding 0.1.

**Pathway enrichment and disease-association analysis.** We used GeneAnalytics, an online universal gene functional analysis tool, which contained more than a hundred data sources for pathway enrichment analysis[95]. Significant DEGs were converted to human gene orthologs semi-automatically. Pathways with prominent functional implications, including neurovascular regulatory pathways, immune/cytokine signaling pathways, respiratory electron transport chain/ATP synthesis, and glucose/energy metabolism pathways were identified from literature review and summarized. We merged enrichment results on TGF-β and VEGF signaling from different sources (i.e. GeneCards and Wikipathways) and presented only the most significant item in the main text/figure.

For neurovascular or neurodegenerative diseases examined (Supplementary Table 2), we compiled GWAS-identified genes associated with disease risk or phenotypic traits from the GWAS catalog and selected recent GWAS studies (see Supplementary Table 2 for full lists of diseases, associated genes and sources). We then found the intersection of the human orthologs of significant DEGs for each cell type/subtype-disease combination and performed hypergeometric test for overrepresentation by the *phyper* function in R.

**Analysis of GLP-1R agonist treatment group scRNA-seq data.** Additional single-cell RNA-sequencing data from the GLP-1R agonist treatment group were processed in parallel with young adult and aged mouse groups by the same pipeline. From the treatment group, we obtained 7239 brain ECs in total, with the following subtype distribution: aEC1, $n = 978$; aEC2, $n = 1155$; capEC, $n = 1728$; vcapEC, $n = 1851$; vEC, $n = 1043$; avEC, $n = 484$. For a given cell type/subtype, the FDR-adjusted P-values and *lnFC* values of upregulated and downregulated genes were calculated for aged versus young group, and exenatide-treated versus untreated aged group. All genes with FDR-adjusted P-value < 0.05 in the aged versus young group comparison were then used to calculate reversal effect. Linear regression was performed and the slope of best linear fit, $r^2$ and P-value for the slope were obtained. Selection of BBB-regulatory, immune/cytokine signaling, and energy metabolism pathway-associated genes were based on annotations from pathway enrichment analysis by GeneAnalytics. For capEC, the proportions of all upregulated and downregulated genes for exenatide-treated versus untreated aged group regardless of statistical significance (i.e. $P_{evo\_up}$ and $P_{evo\_down}$ for DEGs regardless of P-value) were also calculated. Background chance of downregulation of ageing-upregulated statistically significant DEGs by exenatide treatment was $P_{evo\_down}$, while that of upregulation of ageing-downregulated statistically significant DEGs by exenatide treatment was $P_{evo\_up}$. One-sided, one-proportion z-test was used to assay whether the proportions of genes with reversal of expression changes were higher than chance levels.

**Human brain bulk RNA sequencing dataset analysis.** EC-enriched genes were defined as genes with *lnFC* > 0.7 (i.e. at least twofold higher expression) and FDR-adjusted P-value < 0.05 in ECs relative to all other cell types from the mouse brain scRNA-seq data. EC-enriched mouse DEGs were then defined as aged mouse brain EC DEGs (with subtype classification) that are also EC-enriched. Human brain bulk RNA sequencing data (de-identified) from different age groups were obtained from the Genotype-Tissue Expression (GTEx) portal, excluding data from the

cerebellum and cervical spinal cord. In all, 362 samples from 61 subjects were divided into two age groups (<60 and ≥60 years old, from dataset v7). Comparisons of expression levels (expressed in TPM) for all EC-enriched mouse DEGs' human orthologs were carried out across the two age groups by two-sided unpaired t-test with FDR-adjustment for multiple comparisons (with respect to the number of compared genes). Additional analysis by linear regression of expression level versus age was carried out (with exact age, 388 samples from 61 subjects, and dataset v8) and the slope, $r^2$ with associated P-values were obtained. For the Allen Brain Institute Aging, Dementia and Traumatic Brain Injury study dataset (Adult Changes in Thought (ACT) study cohort)[54,55] of human brain bulk RNA sequencing from the Allen Brain Institute, 111 samples from 30 patients were defined as controls with the following criteria: (1) Annotated as "no dementia" based on the DSM IV diagnostic criteria. (2) No history of traumatic brain injury. In total, 58 samples from 16 patients were classified as human AD brain samples with the following criteria: (1) Diagnosed with AD dementia based on the DSM IV diagnostic criteria. (2) No history of traumatic brain injury. All AD subjects included had pathologically confirmed AD with Braak Staging: I, $n = 2$; II, $n = 2$; III, $n = 1$; IV, $n = 2$; V, $n = 5$; and VI, $n = 4$. Comparisons of human brain expression levels (expressed in FPKM) for all EC-enriched mouse DEGs' human orthologs were carried out across the two groups by two-sided unpaired t-test with FDR-adjustment for multiple comparisons (with respect to the number of compared genes). The Mayo Clinic AD brain bulk RNA-seq dataset (the Mayo RNAseq study)[61] downloaded included temporal cortex RNA-seq data from 160 subjects with the following clinical characteristics: 82 AD subjects diagnosed by the NINCDA-ADRDA criteria with Braak Staging of IV or above, and 78 control subjects. For this dataset, the DEGs' expression change magnitudes were retrieved from the publicly available deposited data[96], while FDR-adjusted P-values for the box plots by two-sided unpaired t-test were similarly calculated.

**Mouse brain endothelial cell isolation by immunopanning**. Brain endothelial cells from young and aged mice were purified by immunopanning[97] for further quantitative PCR and western blot assays. Mouse brains ($n = 3$ from each group for each experiment) were dissected, pooled, and diced into ~1 mm³ pieces. Tissues were dissociated into single cells by Neural Tissue Dissociation Kit (P; 130-092-628, Miltenyi Biotec, US). The cells were filtered with 70-µm cell strainer (#352350, Falcon, US) and were further treated with myelin removal beads (130-096, Miltenyi, Germany) to deplete myelin debris and myelin-associated cells. For immunopanning, rat anti-mouse CD45 (550539, BD Pharmingen, US, 1:1500 dilution), and rat anti-mouse CD31 (553370, BD Pharmingen, US, 1:250 dilution) were employed for negative and positive selection respectively. Cells adhered to positive selection dishes were dissociated by trypsinization and harvested.

**Quantitative PCR**. Endothelial cells isolated by immunopanning were lysed immediately in Buffer RLT and subjected to total RNA extraction by the RNeasy® Mini Kit (74104, Qiagen, Germany). Extracted RNA was converted to cDNA by PrimeScript™ RT Master Mix (RR036A, Takara, Japan). Quantitative PCR was performed on a QuanStudio™ 12 K Flex Real-Time PCR system with PowerUp™ SYBR™ Green Master Mix (Applied Biosystems, US). The primers and thermocycling conditions used are listed in Supplementary Table 5. The specificity of all primers were validated by gel electrophoresis. The determination of each gene's cycle threshold ($C_T$) was performed in duplicates. The $C_T$ of each gene was normalized with respect to the $C_T$ of the housekeeping gene Gapdh in the cells of the same group. The values obtained for each gene were then divided by the young adult group mean to obtain the fold change, while two-sided unpaired t-test was employed for inter-group comparisons.

**Western blot**. Endothelial cells isolated by immunopanning were lysed immediately in RIPA buffer and total soluble proteins were quantitated by the Pierce™ BCA Protein Assay Kit (23225, Thermo Fisher Scientific, US). After SDS-PAGE electrophoresis, western blot assays were performed using anti-MFSD2A (PA521049, Thermo Fisher Scientific, US, 1:1000 dilution), anti-SMAD7 (sc-101152, Santa Cruz Biotech, US, 1:200 dilution), anti-LEF1 (sc-374522, Santa Cruz Biotech, US, 1:200 dilution), anti-SLC2A1 (ab652, Abcam, UK, 1:200 dilution), and anti-GAPDH (ab8245, Abcam, UK, 1:2500 dilution) antibodies.

**RNA fluorescent in situ hybridization**. Tissues for RNA FISH were freshly prepared from young and aged mice as specified above. All reagents used were RNase-free and all stocks were made using DNase-/RNase-free water (Invitrogen™, Thermo Fisher Scientific, US). FISH probes were designed using OligoMiner[98] with a specified hybridization temperature of 37 °C and formamide concentration of 50% v/v, 3′-appended with an I2 initiator sequence for hairpin chain reaction[99] plus a TATA, TTTT, or ATAT spacer sequence, and custom-synthesized (Integrated DNA Technologies, Inc., US, or General Biosystems, Inc., US). The probe sequences are listed in Supplementary Table 6.

To perform FISH, mice were deeply anaesthetized and perfused transcardially with 20 ml of ice-cold PBS. Mice were then rapidly decapitated, and whole brains (except the cerebellum) were immersed in OCT compound, snap-frozen in liquid nitrogen, and stored at −80 °C before use. Brains were cut into 10-µm-thick cryosections and collected on coverslips functionalized with (3-aminopropyl)

triethoxysilane. We followed an optimized protocol for tissue treatment previously reported[100] until the probe hybridization step. We then performed probe hybridization and hairpin chain reaction for signal amplification as described in the Mouse Embryo protocol[99] with slight modifications: (1) adding SuperaseIn RNase Inhibitor at 1:100 dilutions for overnight incubation steps, (2) using 4 pmol of each probe for hybridization, and (3) post-staining tissue sections with Hoechst 33342 at 0.1 mg/ml and Dylight® 649-labeled Lycopersicon esculentum lectin (Vector Laboratories, US) at 1:100 dilution for 30 min at room temperature. The washed sections were then immediately imaged without mounting under a Leica TCS SP8 Confocal microscope with a ×40 oil immersion objective (HC PL APO CS2 40x/ 1.30 OIL). The imaging conditions were controlled for the same probe set for young adult and aged mouse brain sections.

**Immunohistochemistry and microglia density analysis**. Mice were deeply anaesthetized and perfusion-fixed transcardially with 10 ml of 4% paraformaldehyde in PBS. Whole brains were harvested and post-fixed at 4 °C overnight. The tissues were sunk in 30% sucrose at 4 °C, embedded in OCT compound, and frozen in chilled 2-methylbutane (isopentane) in a steel beaker using liquid nitrogen prior to storing at −80 °C. The brains were then cryosectioned into 20-µm-thick sections, collected on Superfrost™ Microscope Slides (Invitrogen™, Thermo Fisher Scientific, US), and stored dry at −20 °C until use.

To perform immunohistochemistry, tissue sections were washed in PBS and blocked with normal serum. Anti-IBA1 (019-19741, FUJIFILM Wako Pure Chemical Corporation, Japan) primary antibody at 1:500 dilution and Alexa Fluor 488-conjugated anti-rabbit (A-21206, Invitrogen™, Thermo Fisher Scientific, US) secondary antibody at 1:200 dilution were used for microglia staining. The stained tissues were then mounted with ProLong™ Gold Antifade Mountant with DAPI (Invitrogen™, Thermo Fisher Scientific, US), coverslipped with #1.5 (0.16–0.19 mm) cover glass, and imaged under a Leica TCS SP8 Confocal microscope at field of views (FOV) of $581.25 \times 581.25$ µm² and $290.4 \times 290.4$ µm². Microglia density was analyzed for each imaged region using the Fiji (ImageJ) software with cell counter plugin (64-bit version based on ImageJ 1.52p)[101]. A cell was counted only if the majority of soma was visualized clearly on the imaging plane. The analysis was carried out by one author (Z.L.) and independently verified by another individual (H.K.). The one-way ANOVA test was employed for inter-group comparisons.

**Microglia differential expression analysis**. The analysis pipeline for calculation of DEGs for microglia was similar to ECs as mentioned above. The lnFC and FDR-adjusted P-values of DEGs between aged versus young adult groups, and exenatide-treated versus untreated aged groups were calculated. Known ageing- and neurodegenerative disease-associated genes in microglia examined were selected based on past literature[70,71].

**Cranial window implantation**. To permit in vivo two-photon microscopic imaging of the brain, animals underwent a cranial window implantation surgery. Prior to surgery, mice received one dose of dexamethasone (2 mg/kg, S.C.) to reduce inflammation. Animals were anaesthetized with ketamine (100 mg/kg, I.P.) and xylazine (10 mg/kg, I.P.). Body temperature was maintained at 37 °C by a homeothermic heating pad system. During the surgery, mice were fixed in a stereotaxic instrument (New standard™ 51500D, Parkland Scientific, US). A cranial window (5 mm diameter) was made over the left somatosensory cortex. A customized chamber frame was placed around the opened skull and fixed with cyanoacrylate gel superglue (Loctite 454). The exposed cortex was covered with a 5-mm glass coverslip fixed by cyanoacrylate gel superglue. The rest of the cranial window margin and skull area was filled with dental cement. All procedures were performed carefully to prevent damage to the cortex and its vascular structures. Following surgery, mice were given one dose of enrofloxacin (Baytril, 5 mg/kg S.C.) for prevention of infection and buprenorphine (Temgesic, 0.1 mg/kg S.C.) for analgesia.

**In vivo two-photon imaging assay of BBB integrity**. To assay BBB permeability, mice were anaesthetized with intraperitoneal ketamine (100 mg/kg) and xylazine (10 mg/kg) and placed on a head-fixing apparatus under a custom-modified two-photon microscope (Scientifica, Uckfield, UK) with a Ti:sapphire femtosecond laser (Mai Tai DeepSee, Spectra-Physics, US), galvo-resonant scanners and a Nikon ×16/ 0.80NA water-immersion objective. A dye mixture containing 12.5 mg/ml fluorescein isothiocyanate (FITC)-conjugated dextran (70 kDa molecular weight, Sigma-Aldrich, US) and 6.25 mg/ml tetramethylrhodamine (TRITC)-conjugated dextran (40 kDa molecular weight, Thermo Fisher Scientific, US) dissolved in PBS was administered (4 ml/kg body weight, I.V.) via tail vein injection. In vivo image stacks were then acquired 15–35 min post-injection, with 930 nm excitation wavelength, while the emitted fluorescent signals were detected by photomultiplier tubes through a 520/20-m band-pass filter for FITC-conjugated dextran a 609/34-nm band-pass filter cube for TRITC-conjugated dextran. Laser power under the objective was kept under 50 mW for all imaging sessions. Three image stacks for each mouse were acquired in custom-modified ScanImage (v5.3.1, Vidrio Technologies, US) in MATLAB R2017b (Mathworks, US) and used for the quantification of TRITC-conjugated dextran extravasation. Each image stack had a field of view of 417 µm × 417 µm and consisted of 300 imaging planes evenly distributed

over 239 μm, starting from 50 μm below the cortical surface. 6 images were acquired and averaged for each plane. Measurement of fluorescent signals and volumetric reconstruction of extravasated TRITC-conjugated dextran was carried out using the Imaris software (v6.4, Bitplane, Belfast, UK). To separate intra- and extravascular spaces, volumetric reconstruction of the vasculature was performed using the FITC-conjugated dextran images (i.e. large molecular weight 70 kDa dextran which remained in vessels). Volumetric reconstruction of the TRITC-conjugated dextran in extravascular spaces was then performed and quantified. Volume rendering was performed in normal shading mode. Artefacts were removed by applying gaussian filter. During 3D reconstruction, surfaces of vessels were generated by background subtraction with maximum surface detail limited to 0.814 μm. The FITC-conjugated dextran image stacks were also used to trace the cortical vessels. In brief, a new mask channel was generated by setting the pixel intensity of extravascular and intravascular spaces to 0 and 500 respectively. Filaments were calculated on the new channel with the *Threshold(loop)* algorithm of Imaris with diameter set to 2 μm. The ratio of branch length to trunk radius was set to 3. Default values were adopted for all other parameters. After manually trimming erroneous tracing, the total length of vessels was normalized to the stack volume to obtain vascular length density in unit of μm per mm³. Differences between experimental groups were compared by one-way ANOVA followed by Tukey's multiple comparison test in the Prism software (v6, GraphPad Software Inc., US).

**Reporting summary**. Further information on research design is available in the Nature Research Reporting Summary linked to this article.

## Data availability

The raw and processed RNA sequencing data from this study have been deposited in the UCSC Cell Browser ("Aging Brain [http://cells.ucsc.edu/?ds=aging-brain]"), the Broad Institute Single Cell Portal ("Aging_mouse_brain_kolab [https://singlecell.broadinstitute.org/single_cell/study/SCP829/aging-mouse-brain-kolab]") and the NCBI Gene Expression Omnibus database (accession number: GSE147693). For human aged brain transcriptome comparative analysis, the GTEx datasets were downloaded from [https://www.gtexportal.org/home/datasets] (GTEx_Analysis_2016-01-15_v7_RNASeQCv1.1.8_gene_tpm.gct.gz and GTEx_Analysis_2017-06-05_v8_RNASeQCv1.1.9_gene_tpm.gct.gz) alongside subject information. For the Allen Brain Aging, Dementia, and TBI dataset, the normalized FPKM data matrix was downloaded from [https://aging.brain-map.org/download/index] alongside subject information. For the Mayo Clinic AD brain RNA-seq dataset and DEGs, data was downloaded from [https://doi.org/10.7303/syn3163039] alongside subject information. Source data (underlying Figs. 2d, e and 4b, c, and Supplementary Figs. 7b and 9b) are provided with this paper. Source data are provided with this paper.

## Code availability

R codes for sequencing data analysis are available at GitHub ([https://github.com/RichardLZQ/NVU_code]).

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

## Acknowledgements

We thank members of the Ko laboratory for helpful discussions; Rossa Chiu and Dennis Lo for the generous support and access to sequencing facilities, David Attwell for insightful advice on an earlier version of the manuscript, and William Wu for helpful discussions and access to confocal microscopy; Becky Yung, Anki Miu, Rebecca Chau, Pauline Kwan, and Rachel Hui for providing administrative support to the project. This work was in part funded by the National Key Research and Development Program of China (2016YFC1300600) (V.C.T.M.); a Faculty Innovation Award (FIA2017/B/01) from the Faculty of Medicine, the Chinese University of Hong Kong (CUHK), and a Croucher Innovation Award from the Croucher Foundation (H.K.); the Gerald Choa Neuroscience Center, Margaret K. L. Cheung Research Centre for Parkinsonism Management, Faculty of Medicine, CUHK (V.C.T.M. and H.K.); the Collaborative Research Fund

(C6027-19GF) and the Area of Excellence Scheme (AoE/M-604/16) of the University Grants Committee, Hong Kong (H.K.).

## Author contributions

L.Z., Z.L., and J.S.L.V. carried out scRNA-seq experiments. Z.L. analyzed the scRNA-seq data with advice from H.K.; L.Z. performed in vivo two-photon imaging with assistance from Z.L. and S.K.H.S.; L.Z. and Z.L. analyzed the in vivo imaging data with input from L.Y.C.Y.; Z.L. analyzed human brain transcriptome data with input from H.C.S., W.L.N., and H.K.; X.C. and J.H. performed qPCR and WB validation experiments. H.M.L. performed RNA FISH and reviewed human brain expression databases. L.Y.C.Y. carried out IHC staining. Z.L. analyzed the microglia IHC data. H.Y.E.C., H.C.S., Y.T., and W.J.L. contributed to technical discussions and data interpretation. X.T. and Y.H. contributed to in vivo experimental protocol establishment. Z.L. and X.C. prepared manuscript figures. Z.L., H.M.L., V.C.T.M., and H.K. wrote the manuscript with input from all authors.

## Competing interests

Exenatide used in the study was provided by AstraZeneca Hong Kong Limited. L.Y.C.Y. and J.H. were employed by Aptorum Group Limited and honorary research staff of CUHK. AstraZeneca Hong Kong Limited and Aptorum Group Limited had otherwise no role in the funding, design, or execution of the study.
