## [Peer Review File · Nature Communications]

Reviewers' Comments:

Reviewer #1:

Remarks to the Author:

The authors present a report detailing the vascular zone-specific changes in brain endothelial cells during aging that correspond with changes that occur in human aging and Alzheimer's disease. They use single cell RNAseq to dissect the contributions that the different endothelial sub-types make to aging, and also find that these are reversed when treated with exenatide. This work is timely, as there is considerable interest in using single cell technologies to dissect cerebrovascular changes in aging and neurodegeneration (10.1038/s41591-019-0440-4; 10.1038/s41586-019-1362-5). This work effectively uses single cell genomics to characterise changes to the ECs in different segments brain vasculature that occur during aging. The finding that aged BECs have an inflammatory gene expression signal is interesting, and in line with 'inflamm-aging' literature. The finding that BECs have potentially reduced metabolic capacity is also interesting, because their BBB function is highly ATP-dependent. Furthermore, the authors relate EC subtype changes to neurodegeneration risk genes, and find DEGs in capillary ECs are enriched in AD risk genes, implying a role for these EC subtype changes in AD. The key contribution of this paper to the thinking in the field will be in encouraging researchers to consider endothelial zonation when they examine changes to the brain vasculature.

Major comments:

- 1) While interesting, there is a lack of immunohistochemistry or in situ hybridisation to validate single cell RNAseq changes. If some validation experiments were performed it would go a long way to reassuring the reader that these changes were meaningful/functional.
- 2) Comparing bulk RNAseq to single cell work is fraught in disease/aging where cell populations change, accounting for most of the gene expression changes observed. In neurodegenerative diseases, where neurons die off, other cell types will be enriched in relative terms leading to apparent increases in gene expression. This will tend to exaggerate gene expression increases and mask decreases in non-neuronal cells, and may account for discordances. This makes it difficult to compare bulk vs sorted transcriptomes. The strongest evidence that age-related changes in BECs are related to human AD is in F3, and maybe F4 might be better in the supplementary material.
- 3) The rationale behind targeting GLP1R seems lacking, and could be explained better, leveraging the scRNAseq data. Why would you expect exenatide to affect BEC function? Where is GLP1R expressed? Does exenatide directly target BECs? Are there changes in the BECs that would indicate a deficiency in GLP1R signalling to justify using exenatide? Addressing these questions would help justify/rationalise why exenatide may be affecting the brain vasculature. Alternatively, the authors may wish to hold on to this until they have a convincing mechanism, and publish this separately.
- 4) The authors claim to provide the first evidence of zonation-specific endothelial changes in aging, however they could further acknowledge a similar study by Yousef et al. (10.1038/s41591-019-0440-4). It might be helpful to reference this in the intro as well, and the parallels between the studies. Yousef et al. however, highlight the role that venous/arterial ECs play in immune cell tethering during aging – rather than the role of capillary bed ECs, which is highlighted here. Do the authors also see VCAM1 induction in arterial/venous EC populations? It would be worth mentioning, either way. It is interesting to note that neither paper saw leakage of 70 kDa dextran, but Yousef et al. did not test lower MW dextrans, which were shown to extravasate here.
- 5) Would the authors consider creating a searchable database/website for non-experts to easily access these data?

Minor points:

- 1) Could the preponderance of capillary EC changes be proportional to the oversampling of this population? This would provide an alternative interpretation, as statistical tests would have more power to detect changes, as opposed to more severe changes to the capillary bed.
- 2) The single-cell RNAseq work is convincing, and provides more thoroughly delineated EC

- zonation than Vanlandevijck et al. (<https://doi.org/10.1038/nature25739>) or Yousef et al. (10.1038/s41591-019-0440-4). It may however be important to consider the paper in the broader context of single cell/EC genomic studies finding changes to ECs in the aging/diseased brain: Tabula Muris Senis (10.1101/661728), Dulken et al. (10.1038/s41586-019-1362-5), Munji et al. (10.1038/s41593-019-0497-x). Also studies that characterise the brain at single cell resolution: Tabula Muris (10.1038/s41586-018-0590-4), DropViz (10.1016/j.cell.2018.07.028).
- 3) In F4, p8, is FDR adjusted for the number of genes you analysed? Or for the entire transcriptome?
 - 4) In F4, would it not be better to run regressions of age vs GE, rather than artificially dividing into two groups?
 - 5) In F4, would the authors also find the publicly available Mayo Clinic AD brain bulk RNAseq dataset (10.1038/sdata.2016.89) useful?
 - 6) P11: "Among BBB-associated pathways with enrichment of expression changes, altered TGF- β , VEGF and immune/cytokine signalling appear to impact all vascular segments, while others exhibit subtype-specific changes that notably all impact capEC." Ref: 10.1186/s12974-018-1167-8.
 - 7) P12: Paragraph 2, start of third line, typo, Intriguing – should be intriguingly.
 - 8) P12: Paragraph 2, end of fifth line. GLUT1-deficient or heterozygous GLUT1 deletion mice not GLUT1 knockdown.
 - 9) P15: Minor concern, but saline or another type of vehicle is not used to control the exenatide experiments.
 - 10) P16: Second to last line, typo, FindNerighbors.
 - 11) S2a: subtype-biased?

Reviewer #2:

Remarks to the Author:

This study essentially comprises 3 comparisons: (i) young vs aged mouse endothelium, (ii) aged vs human AD database, and (iii) aged mouse endothelium vs exenatide-treated mouse endothelium. The overall direction of this study is interesting but there are major issues that must be carefully addressed.

1. The conclusions and focus appear to be on BBB. For example, on page 6 ... "Overall, capEC had the most prominent BBB-regulatory signaling pathways enrichment". However, many of the genes and signaling pathways identified also subserved any other critical non-BBB functions, e.g. TGF, VEGF and immune/cytokine signaling. It is not clear whether the singular focus on the BBB is justified.
2. BBB regulation is a complex process, and there is not just a "linear" relationship between so-called "BBB genes" and BBB function. For example, the authors also found that several genes with important functions at the BBB such as *Cldn5*, *Slc2a1*, *Ifitm3*, had increased expression in aged human brains but downregulated in aged mouse brain ECs (Fig 4a).
3. Although GLP-1R agonist treatment reverses aging-associated EC transcriptomic changes and reduces BBB leakage and the treatment also generalizes to the subcategory of DEGs whose human orthologs are AD-associated (Fig 5e) in capEC, it is still hard to get the conclusion that BBB protection is an important underlying mechanism that explains GLP-1R agonists' benefits in AD. In fact, some studies suggest that there may be limited uptake of therapeutic antibodies in multiple mouse models of AD, perhaps owing to intact BBB and limited permeability (Bien-Ly et al, Neuron 2015).
4. In Fig 4, the authors identified similar expression changes in normal aged mouse and human AD brains. The changes of some EC-enriched genes in aged mouse brain, had similar direction with AD human brain data, but not the same with aged human brain data, e.g. *Slc2a1*. But aging is not the same as AD. What is the possible meaning of this? Does this mean the normal aged mouse can

model human AD? Are there any genes show concordant changes in aged mouse and aged human brain, but different in AD human brain? These might be more related with AD pathology.

5. The rather sudden "jump" to exenatide is unclear. Not much rationale is provided for GLP-1R. Why choose this target? It is mentioned that GLP-1R agonists have been reported to provide neuronal protection, but is there also some justification for selecting this for BBB and endothelium? In fact, GLP-1R may be a universally expressed gene. Why this focus in aged mouse brain, or aged or AD human brains?

6. The cell number of endothelial cells from 5 aged mice appear to be lower than the one from 5 young mice. It seems unclear whether this was due to a batch effect, or just because aged brains have a fewer number of endothelial cells. Vascular density may be important when calculating dextran data? When the authors using dextran to detect permeability of BBB, did they check the density of the vessels? It seems the vessels density might be higher in the aged mouse's cortex than the young mouse?

7. The authors may need to clarify whether they prepared single-cell RNAseq libraries from mice that were also used for two-photon imaging studies. The surgery for cranial window (and the procedures of two-photon imaging) may change the mRNA expression patterns in brain cells.

8. In Fig 5b: they stated "n=9 image stacks from 3 mice for each group". Does this mean that the values for mean +/- SEM in Fig 5b were from n=9? If yes, probably, they need to recalculate the average (and SEM) from 3 mice (the FC value for each mouse should be from the average FC value of 3 image stacks in each mouse?).

9. The authors should give a confirmation on cell purity: high gene expression values for genes with EC-specific expression and very low or undetectable expression for other cell- type-specific markers of the central nervous system.

10. Some key signals should be confirmed with protein.

11. The authors used whole brain to prepare the libraries, but did not discuss the brain-region-related heterogeneity of the data. Especially because aging affects specific brain areas differently, the authors might wish to address whether region-specific (such as cortices and hippocampi etc) display key differences in gene expression patterns upon aging.

12. If space allows, it may be nice to provide some discussion into this point. It may be nice to discuss potential sex differences because this study used only male mice.

13. Finally, how "specific" are the endothelial patterns found in this study? There are many other papers that have previously mapped aging transcriptomes for brain endothelium, astrocytes, microglia etc (e.g. Guo et al, *Neurobiol Dis* 2019; Boisvert et al, *Cell Rep* 2018; Olah et al, *Nat Comm* 2018). It is a bit surprising that these previous papers were not referenced or discussed.

We thank the reviewers for their constructive suggestions and criticism. We have included more data to address these concerns, and made changes to parts of the manuscript where we or the reviewers felt more clarity or detail was required. Major additions and changes include:

- New validation experiments for the expression changes reported, whereby we supplemented scRNA-seq with RNA fluorescent *in situ* hybridization to confirm brain endothelial expression of selected genes in both aged groups (Supplementary Fig. 4), as well as quantitative PCR and western blot in isolated brain endothelial cells from the different age groups to verify the differential expressions for a subset of the genes at both the transcript and the protein levels (Fig. 2c-e).
- Improved background coverage and discussion to explain our choice of testing GLP-1R agonist treatment. We also provided additional information on the expression pattern of GLP-1R in the brain (Rebuttal Fig. 1), in relation to mechanistic speculations.
- Restructuring of the paper, with the results on comparative analysis to aged human brain bulk RNA-seq dataset moved to supplementary information (Supplementary Fig. 6), and acknowledging the precaution on result interpretation. The concordant downregulation of several genes important at the BBB in the human AD brain are now presented alongside the enrichment of AD GWAS genes in capillary EC (capEC) DEGs (Fig. 3), and supplemented with additional comparative analysis utilizing the Mayo Clinic AD brain bulk RNA-seq dataset (Supplementary Fig. 5).
- Additional citations and discussions of our findings in relation to numerous references suggested by the reviewers. We supplemented additional analysis on the expression levels of Vcam1 in arterial/venous EC (avEC) in ageing in our dataset (Supplementary Fig. 3), and compared our findings to that of Yousef et al. (2019).
- Additional experimental results and analysis to extend functional measures assayed in relation to aged brain EC transcriptomic changes found, whereby cortical vascular length density (Fig. 4c), IBA1+ microglia density, microglial upregulation of ageing-, neurodegenerative disease- and activation-associated genes and their potential influence by GLP-1R agonist treatment have now been examined (Supplementary Fig. 9).

All the points are addressed individually in detail below.

Reviewer #1 (Remarks to the Author):

The authors present a report detailing the vascular zone-specific changes in brain endothelial cells during aging that correspond with changes that occur in human aging and Alzheimer's disease. They use single cell RNAseq to dissect the contributions that the different endothelial sub-types make to aging, and also find that these are reversed when treated with exenatide. This work is timely, as there is considerable interest in using single cell technologies to dissect cerebrovascular changes in aging and neurodegeneration (10.1038/s41591-019-0440-4; 10.1038/s41586-019-1362-5). This work effectively uses single cell genomics to characterise changes to the ECs in different segments brain vasculature that occur during aging. The finding that aged BECs have an inflammatory gene expression signal is interesting, and in line with 'inflamm-aging' literature. The finding that BECs have potentially reduced metabolic capacity is also interesting, because their BBB function is highly ATP-dependent. Furthermore, the authors relate EC subtype changes to neurodegeneration risk genes, and find DEGs in capillary ECs are enriched in AD risk genes, implying a role for these EC subtype changes in AD. The key contribution of this paper to the

thinking in the field will be in encouraging researchers to consider endothelial zonation when they examine changes to the brain vasculature.

We thank the reviewer for the encouraging remarks that our study is timely, and the findings are interesting. We are also grateful that the reviewer has made very important comments and constructive criticisms to help us improve the study. We have now substantially revised the manuscript to address the concerns, by additional experimentation, analyses and revision of the writings. Our detailed point-by-point responses to each of the comments are as follows.

Major comments:

1) While interesting, there is a lack of immunohistochemistry or in situ hybridisation to validate single cell RNAseq changes. If some validation experiments were performed it would go a long way to reassuring the reader that these changes were meaningful/functional.

We thank the reviewer for the advice and we completely agree that further validation experiments will help us consolidate the findings. We have now carried out RNA fluorescent *in situ* hybridization (FISH) and immunohistochemistry (IHC) for selected genes involved in key regulatory pathways (Supplementary Fig. 4) in brain slices from both age groups (including *Afdn*, *Lef1*, *Ptprb*, *Mfsd2a*, *Smad7*, *Flt1*, *Ifitm3*, *Jund*, *Slc9a3r2* and *Nostrin*). As the main comparison in the study was made between young adult vs normally aged mice, most differential expressions detected were not of large magnitude and we found convincing quantitative comparisons based on the images obtained difficult (Supplementary Fig. 4). We reasoned that while RNA FISH and IHC can verify vascular localization and endothelial expression of individual genes, quantitative validation of differential expression necessitates more sensitive methods by qPCR and western blot (WB).

We thus additionally carried out qPCR and WB, and validated the differential expression of selected genes involved in key pathways or regulation of blood-brain-barrier (BBB) functions (e.g. adherens junction, VEGF and TGF- β signalling-related genes, key transporter genes) in pooled endothelial cells (ECs) isolated from whole mouse brains by immunopanning (Fig. 2c, also see Methods). With qPCR, we were able to verify the altered expression of several genes involved in key regulatory pathways of neurovascular functions (e.g. VEGF and TGF- β signalling-related genes, key transporter genes), including both aged brain EC-upregulated (*Flt1*, *Klf6*, *Smad7*) and downregulated (*Mfsd2a*) genes (Fig. 2d) For protein targets for which we identified suitable antibodies (*LEF1*, *SMAD7* and *MFSD2A*), we also validated the expression changes at the protein level by WB (Fig. 2e). These results are now incorporated in the revised manuscript (page 6, paragraph 3).

2) Comparing bulk RNAseq to single cell work is fraught in disease/aging where cell populations change, accounting for most of the gene expression changes observed. In neurodegenerative diseases, where neurons die off, other cell types will be enriched in relative terms leading to apparent increases in gene expression. This will tend to exaggerate gene expression increases and mask decreases in non-neuronal cells, and may account for discordances. This makes it difficult to compare bulk vs sorted transcriptomes. The strongest evidence that age-related changes in BECs are related to human AD is in F3, and maybe F4 might be better in the supplementary material.

We are grateful to the reviewer for this very important advice on analysis and the positive remark that the data presented in Fig. 3 provide strong evidence for a relationship between age-related changes in brain ECs and human AD. We agree that results from the comparative analysis must be interpreted with caution. As per the reviewer's suggestion, we tuned down our findings from this part of the study and moved the comparative analysis with normal aged human brain bulk RNA-seq data in the original Fig. 4 to supplementary information (now as Supplementary Fig. 6). We also acknowledge the precaution on data interpretation in the main text (page 8, paragraph 2).

On the other hand, we still observed concordant downregulated expression of several EC-enriched genes in human AD brains from the Allen Brain Ageing, Dementia and TBI dataset. As the downregulation of these non-neuronal, EC-enriched genes would otherwise tend to be masked for the same consideration, the fact that we still found a significant downregulation may imply that their expression changes are more prominent in the human AD brains. We therefore have kept this part in the main results (Fig. 3b, c). Furthermore, for *Mfsd2a*, we have carried out additional validation of differential expression by qPCR and WB in isolated brain ECs from mice of the two age groups (Fig. 2c-e) to strengthen our conclusion.

3) The rationale behind targeting GLP1R seems lacking, and could be explained better, leveraging the scRNAseq data. Why would you expect exenatide to affect BEC function? Where is GLP1R expressed? Does exenatide directly target BECs? Are there changes in the BECs that would indicate a deficiency in GLP1R signalling to justify using exenatide? Addressing these questions would help justify/rationalise why exenatide may be affecting the brain vasculature. Alternatively, the authors may wish to hold on to this until they have a convincing mechanism, and publish this separately.

We apologize for our explanations on the rationale of testing the efficacy of an GLP-1R agonist were inadequate. In the revised manuscript, we have now explained in more detail the heuristics behind our choice. In short, we made the following observations: (1) human clinical trials have found potential efficacies of GLP-1R agonists in two neurodegenerative conditions, improving brain metabolism in AD and delaying motor deterioration in Parkinson's disease (PD) (refs 39 and 40: Gejl et al. (2016), Athauda et al. (2017)), (2) in animal experimentation, GLP-1R agonists were found to be efficacious in ameliorating the progression of neuropathology in animal models of AD, PD, amyotrophic lateral sclerosis and stroke (refs 33-38: Kim et al. (2009), Li et al. (2009, 2010, 2012), McClean et al. (2011), Yun et al. (2018)), (3) the use of GLP-1R agonists in diabetic patients reduce the incidence of cardiovascular events (e.g. myocardial infarction, stroke) (reviewed in ref 41: Kristensen et al. (2019)), while animal tests revealed vascular protective effects in peripheral organs (ref 42 and 43: Noyan-Ashraf et al. (2009), Helmstadter et al. (2019)), and (4) we observed that impaired energy metabolism and altered immune/cytokine signalling were implicated especially in capillary ECs by the scRNA-seq data. At other body sites (e.g. the myocardium), GLP-1R signalling has been reported to alter energy metabolism and exert immunomodulatory effects (refs 44-48: Arakawa et al. (2010), Yusta et al. (2015), Bruen et al. (2017), Vinue et al. (2017), Filippidou et al. (2020)). With these considerations, we thus postulated that GLP-1R agonist may have neurovascular protective effects that could help explain its general applicability as potential therapeutics for multiple neurodegenerative conditions with age-associated neurovascular defects as a shared pathological component. These points are also now discussed in the Introduction (page 3, paragraph 1) and Results sections (page 9, paragraph 1).

The expression pattern of GLP-1R in the brain is still an active area of research, whereby past literature has revealed its presence in subsets of neurons, microglia, while peripherally it may be expressed in vascular cells (refs 37, 49-51: Yun et al. (2018), Cork et al. (2015), Lovshin et al. (2015), Jensen et al. (2018)). We have examined the expression pattern of GLP-1R in the mouse brain by RNA FISH ourselves, apart from referencing previous literature (page 3, paragraph 1). Note that in both our scRNA-seq dataset and that by Vanlandewijck, et al. (2018), we did not find GLP-1R transcripts in ECs. RNA FISH revealed almost no localization of GLP-1R transcripts to ECs (please see attached Rebuttal Fig. 1 at the end of this response letter). It thus appeared that ECs had low expression of GLP-1R, if any. We have discussed the potential mechanisms that may underlie the transcriptomic reversal and BBB protective effects of GLP-1R agonist we found (page 13, paragraph 2). Indeed, there are multiple possible underlying mechanisms of action, whereby we postulate that indirect actions via microglia or modulation of compositions in the peripheral circulation are important to consider. This is also supported by our observation that the upregulation of microglial ageing- and neurodegenerative disease-associated transcripts was decreased by GLP-1R agonist treatment (Supplementary Fig. 9). As the reviewer kindly suggested, we wish to hold on to this and publish separately when we have convincingly identified the exact mechanism(s).

4) The authors claim to provide the first evidence of zonation-specific endothelial changes in aging, however they could further acknowledge a similar study by Yousef et al. (10.1038/s41591-019-0440-4). It might be helpful to reference this in the intro as well, and the parallels between the studies. Yousef et al. however, highlight the role that venous/arterial ECs play in immune cell tethering during aging – rather than the role of capillary bed ECs, which is highlighted here. Do the authors also see VCAM1 induction in arterial/venous EC populations? It would be worth mentioning, either way. It is interesting to note that neither paper saw leakage of 70 kDa dextran, but Yousef et al. did not test lower MW dextrans, which were shown to extravasate here.

We have now referenced Yousef et al. (2019) in the Introduction (page 2, paragraph 3), and examined the expression levels of *Vcam1* in arterial/venous EC (avEC). Possibly due to a limited avEC cell number in our current dataset, we did not observe a significant differential expression of *Vcam1* in aged brain avEC ($\lnFC = -0.39$; FDR-adjusted P -value = 1; page 6, paragraph 1; Supplementary Fig. 3a). As *Vcam1* is only expressed by a very small proportion of brain ECs, Yousef et al. (2019) reported that comparison of *Vcam1* expression across age by bulk RNAseq of pooled brain ECs did not reveal any differences (at age groups similar to that used in our study: 3 and 19 months old; see Extended Data Fig. 1d of Yousef et al. (2019)). They therefore employed scRNA-seq on FACS-sorted CD31+VCam1+ mouse hippocampal ECs and only then found a significant increase in expression in the aged group. To make a similar comparison, we also performed comparison by inclusion of only avECs with non-zero *Vcam1* transcript counts. However, we still did not observe a significant differential expression ($\lnFC = -0.19$; FDR-adjusted P -value = 1; Supplementary Fig. 3b). We propose this may have due to (1) regional heterogeneity of differential expression of *Vcam1*, whereby Yousef et al. (2019) isolated hippocampal CD31+VCam1+ ECs while our dataset was obtained from whole-brain ECs similar to Vanlandewijck et al. (2018), and (2) the differences in protocols adopted for isolation, sequencing and comparison. We have also additionally highlighted the similarity of results on large molecular weight dextran leakage in the Results section (page 9, paragraph 2), and the relationship of our findings of aged brain EC DEG-enrichment of immune/cytokine signalling and that of Yousef et al. (2019) in the Discussion (page 11, paragraph 2).

5) Would the authors consider creating a searchable database/website for non-experts to easily access these data?

We agree that a searchable database would facilitate access of the data by others. Invited by peers from the University of California, Santa Cruz (UCSC) Genomics Center, our data (both raw and processed) is now hosted on the UCSC single-cell RNA sequencing portal (accessible at URL 1 below) and can be interactively accessed using the UCSC Cell Browser for easy data visualization. We have also deposited the data at the Broad Institute single cell portal (accessible at URL 2 below), and the Gene Expression Omnibus (accession number: GSE147693). R codes for detailed analysis will also be provided upon reasonable requests from readers after publication of the manuscript.

URL 1: <http://cells.ucsc.edu/?ds=aging-brain>

URL 2: http://singlecell.broadinstitute.org/single_cell/study/SCP829/aging-mouse-brain-kolab

Minor points:

1) Could the preponderance of capillary EC changes be proportional to the oversampling of this population? This would provide an alternative interpretation, as statistical tests would have more power to detect changes, as opposed to more severe changes to the capillary bed.

We were also cautious with this potential confounding factor when interpreting the results. We therefore analyzed the dependence of significant DEG number on EC-subtype cell number (Supplementary Fig. 2a). For the different EC subtypes, despite that increased statistical power certainly comes with larger cell number, we did not find a simple monotonic relationship (Supplementary Fig. 2a). For instance, although we sampled more venous-capillary EC (vcapEC) than capillary EC (capEC), capEC exhibited substantially more significant differential expressions than vcapEC (Fig. 1 f, g; Supplementary Fig. 2a). Indeed, vcapEC had similar DEG number as arterial ECs with only approximately halved cell numbers (Fig. 1 f, g; Supplementary Fig. 2a). We thus considered the more profound changes in capECs likely reflected more prominent biological changes. For the avEC subtype, we do acknowledge that the small number of significant DEGs we found may be caused by a relatively smaller sample size (also see response to major comment 4 regarding differential expression of Vcam1 in avEC). These points are also discussed in the revised manuscript (page 4, paragraph 2).

2) The single-cell RNAseq work is convincing, and provides more thoroughly delineated EC zonation than Vanlandevijck et al. (10.1038/nature25739) or Yousef et al. (10.1038/s41591-019-0440-4). It may however be important to consider the paper in the broader context of single cell/EC genomic studies finding changes to ECs in the aging/diseased brain: Tabula Muris Senis (10.1101/661728), Dulken et al. (10.1038/s41586-019-1362-5), Munji et al. (10.1038/s41593-019-0497-x). Also studies that characterise the brain at single cell resolution: Tabula Muris (10.1038/s41586-018-0590-4), DropViz (10.1016/j.cell.2018.07.028).

We are thankful that the reviewer considers the scRNA-seq work convincing with clear delineation of EC zonation. In the revised manuscript, we have discussed our findings in relation to the works the reviewer kindly suggested. In brief, we have now referenced the related works of Tabula Muris and Saunders et al. (2018) (DropViz) in the Abstract, which performed large-scale single-cell transcriptomic profiling similar to that by Ziesel et al. (2018) we originally cited. We also related our findings on the enrichment of immune/cytokine signalling-associated genes in the aged

brain EC DEGs to that of Dulken et al. (2019) and Munji et al. (2019), which reported expression changes implicating altered immune cell and cytokine signalling in the subventricular zone in ageing and at the BBB of affected regions in several neurological disease models respectively, in the revised Discussion (page 11, paragraph 2). Finally, Tabula Muris Senis indeed encompasses a tour de force effort in annotating and profiling single-cell transcriptomic alterations in diverse organs across age, while we have only focused specifically on the brain vasculature. This work is also now referenced in the Discussion (page 14, paragraph 1).

3) In F4, p8, is FDR adjusted for the number of genes you analysed? Or for the entire transcriptome?

The FDR adjustment was performed with respect to the number of compared genes, which consisted of all detected genes in the transcriptome. We have now clarified this in the Methods (page 18, paragraph 2).

4) In F4, would it not be better to run regressions of age vs GE, rather than artificially dividing into two groups?

We have performed additional regression analysis. Number of genes found with significant expression changes with respect to age by regression were less than that obtained by binarizing the age groups, yet largely overlapping (18 out of 19 genes with significant expression changes on regression were found significantly differentially expressed when compared by segregation of data into two age groups, i.e. ≥ 60 versus < 60 years old). These are now included in the revised manuscript (page 8, paragraph 2; Supplementary Fig. 6b, d).

5) In F4, would the authors also find the publicly available Mayo Clinic AD brain bulk RNAseq dataset (10.1038/sdata.2016.89) useful?

We thank the reviewer for pointing to this dataset. We have performed additional comparative analysis using the Mayo Clinic AD brain bulk RNAseq dataset. Among the four EC-enriched genes newly found to exhibit concordant expression changes in aged mouse brain and human AD brain based on this analysis, we noted the upregulation of the *DLC1* gene, an AD GWAS gene that has also recently found to be upregulated in the entorhinal cortex ECs of AD patients (Grubman A et al. (2019)), among others. The magnitude of expression changes for the other three genes (*IGF1R*, *ARHGAP29* and *IQGAP1*) were similar or greater than that of *DLC1*.

The difference of genes found with the Allen Brain and Mayo Clinic datasets could possibly reflect the heterogeneity of human AD brain datasets due to difference in sample size, regional sampling, tissue processing and sequencing protocols adopted. We believe these analyses have brought us new insights into possible gene candidates with age-associated expression changes and vascular mechanistic linkages to AD. With the precaution on data interpretation that the upregulation of non-neuronal genes may be exaggerated in human AD brain bulk RNAseq, these findings are now presented in the revised manuscript as supplementary information (page 7, paragraph 2; Supplementary Fig. 5).

6) P11: “Among BBB-associated pathways with enrichment of expression changes, altered TGF- β , VEGF and immune/cytokine signalling appear to impact all vascular segments, while others exhibit subtype-specific changes that notably all impact capEC.” Ref: 10.1186/s12974-018-1167-8.

We have now cited the reference suggested (page 11, paragraph 1).

7) P12: Paragraph 2, start of third line, typo, Intriguing – should be intriguingly.

We have corrected this typo.

8) P12: Paragraph 2, end of fifth line. GLUT1-deficient or heterozygous GLUT1 deletion mice not GLUT1 knockdown.

Thanks for pointing this out, we have now corrected the text.

9) P15: Minor concern, but saline or another type of vehicle is not used to control the exenatide experiments.

For testing the effect of BBB protection by exenatide treatment, we had an additional independent batch of experiments carried out, whereby we verified the BBB-protective effects of exenatide treatment over saline vehicle injection. We however did not perform further sequencing experiments on the vehicle control group. The data is now included as supplementary information (page 9, paragraph 2; Supplementary Fig. 7).

10) P16: Second to last line, typo, FindNerighbors.

We have now corrected this typo.

11) S2a: subtype-biased?

Apologies for the misleading labelling, we meant to express DEGs obtained with EC subtype classification. We have now revised the wording to ensure better clarity.

Reviewer #2 (Remarks to the Author):

This study essentially comprises 3 comparisons: (i) young vs aged mouse endothelium, (ii) aged vs human AD database, and (iii) aged mouse endothelium vs exenatide-treated mouse endothelium. The overall direction of this study is interesting but there are major issues that must be carefully addressed.

We thank the reviewer for the positive remark that the direction of the study is interesting. We also thank the reviewer for pointing out important issues to address and giving us important advice that significantly helped us improve the manuscript. We have carried out additional experiments, performed further analyses and substantially revised the writings to address the concerns. Our detailed point-by-point responses to the issues raised are as follows.

1. The conclusions and focus appear to be on BBB. For example, on page 6 ... “Overall, capEC had the most prominent BBB-regulatory signaling pathways enrichment”. However, many of the genes and signaling pathways identified also subserve any other critical non-BBB functions, e.g. TGF, VEGF and immune/cytokine signaling. It is not clear whether the singular focus on the BBB is justified.

We fully agree with the reviewer that functional changes beyond BBB integrity relevant to the ageing-associated transcriptomic changes found are also worth assayed and discussed. To address this concern, we have revised the manuscript to more precisely reflect the diverse roles of enriched signalling pathways (see point (i) below) and strengthen our discussions of their linkage to BBB integrity (see point (ii) below). Instructed by transcriptomic and BBB changes found in the aged brain ECs, further related measures (see points (iii) and (iv) below) have now been also assayed.

(i) We revised Fig. 2 labelling and legends, to more precisely describe the important pathways with enrichment (e.g. VEGF and TGF- β signalling) as “*Signalling pathways with diverse functions including vascular and BBB regulation*”. We have also revised the main text at appropriate places to ensure consistency and accuracy on this point (page 3, paragraph 2; page 5, paragraph 3; page 6, paragraph 4).

(ii) The endothelial cell (EC) subtype-dependent immune/cytokine signalling-related transcriptomic changes is indeed also an important part of our main results, we have thus revised the Introduction to provide better coverage of the background (page 2, paragraph 3), and strengthened our discussions on the implications on BBB integrity regulation in the Results (page 5, paragraph 3) and Discussion sections (page 11, paragraph 2). Specifically, we propose that as VEGF and the numerous enriched cytokine signalling pathways of aged brain capillary EC and venous-capillary EC (e.g. IL-1, IL-2, IL-6, and IL-17) have been reported to disrupt endothelial barrier integrity *in vitro*, or BBB integrity *in vivo* in various disease models (refs 56-61: Anthony et al. (1997), Blamire et al. (2000), Kebir et al. (2007), Linker et al. (2008), Argaw et al. (2012), Wylezinski et al. (2016)), their appropriate signalling are likely required for BBB integrity. This substantiates our choice to select BBB integrity as a key functional measure.

(iii) As some signalling pathways with significant enrichment in aged brain EC DEGs play important roles in regulating angiogenesis / vasculogenesis (e.g. VEGF and TGF- β signalling), we carried out additional analyses comparing cortical vascular length density across age and treatment groups. We however did not find differences between the groups (page 9, paragraph 2; Fig. 4c; also see response to comment 6).

(iv) The brain endothelium bidirectionally communicates with microglia, whereby one direct consequence of brain EC transcriptomic changes and BBB leakage in ageing is increased microglia activation, which could also reciprocally influence the endothelium (refs 92-96: Nimmerjahn et al. (2005), Roth et al. (2013), Lou et al. (2016), Li et al. (2017), Haruwaka et al. (2019)). We therefore additionally examined aged brain microglia density by IBA1 immunostaining, and the expression levels of ageing-, neurodegenerative disease- and activation-associated microglial genes (refs 98-99: Keren-Shaul et al. (2017), Lloyd et al. (2019)). We found that GLP-1R agonist treatment exhibited a trend of reducing the age-related increase of IBA1+ microglial density (Supplementary Fig. 9a, b). Interestingly, we also found that the upregulation of numerous ageing- and neurodegenerative disease-enriched microglial genes (including Apoe, Ccl4, Ccl6, Cd9, Cd52, Timp2), which may play a role in priming microglia to be activated (refs 98-99: Keren-Shaul et al. (2017), Lloyd et al. (2019)), was attenuated by GLP-1R agonist treatment (Supplementary Fig. 9c, d). For the EC transcriptomic changes found and its reversal by GLP-1R agonist treatment, we have thus further extended associated functional measures assayed beyond BBB to age-related microglial priming.

2. BBB regulation is a complex process, and there is not just a “linear” relationship between so-called “BBB genes” and BBB function. For example, the authors also found that several genes with important functions at the BBB such as Cldn5, Slc2a1, Ifitm3, had increased expression in aged human brains but downregulated in aged mouse brain ECs (Fig 4a).

This is a very important comment which we fully agree with. Currently, we do not have a definite explanation for why certain genes with important functions at the BBB exhibit discordant

expression changes in the aged human brain, nor can we claim that these changes we found could fully account for the altered BBB integrity in ageing. We thus do acknowledge that the interpretation of transcriptomic data is often associative and not causal (page 13, paragraph 4), and that to establish a causal relationship between the expression changes found and altered neurovascular functions, further mechanistic studies involving enhanced expression, knockdown or knockin/out of specific genes(s) are required (page 13, paragraph 4).

Possibly due to a reduction of neurons, human aged brain bulk RNAseq results may bias towards exaggerating upregulation and masking downregulation of non-neuronal genes (a concern raised by reviewer 1). In view of this uncertainty on interpretation which may account for some discordant expression changes, we have moved the results on comparative analysis against human aged brain bulk RNAseq data to the supplementary information (Supplementary Fig. 6). We also discuss the potential caveat in interpretation in the revised main text (page 8, paragraph 2).

3. Although GLP-1R agonist treatment reverses aging-associated EC transcriptomic changes and reduces BBB leakage and the treatment also generalize to the subcategory of DEGs whose human orthologs are AD-associated (Fig 5e) in capEC, it is still hard to get the conclusion that BBB protection is an important underlying mechanism that explains GLP-1R agonists's benefits in AD. In fact, some studies suggest that there may be limited uptake of therapeutic antibodies in multiple mouse models of AD, perhaps owing to intact BBB and limited permeability (Bien-Ly et al, Neuron 2015).

We thank the reviewer for raising this interesting discussion point. We propose that whether improving BBB integrity has beneficial or detrimental roles in the pathogenesis and treatment of AD may be context dependent.

In the vascular two-hit hypothesis of AD pathogenesis, neurovascular dysfunction with BBB breakdown is an early event that increases brain tissue exposure to neurotoxins and promotes neuroinflammation. These processes in turn exacerbate other pathological processes, such as amyloid beta and tauopathy-associated neurotoxicity, to cause synaptic and neuronal degeneration. As ageing is the greatest risk factor of AD and BBB breakdown in ageing contributes to the development of neuropathology, we believe that in this context BBB protection represents a potentially promising therapeutic strategy for preventing or delaying neurodegeneration. The implication from our study is therefore more relevant to early or presymptomatic stages of AD, when rescuing ageing-associated BBB breakdown may be beneficial.

On the other hand, the reviewer has raised an important consideration regarding brain exposure to therapeutic antibodies. It is indeed essential, that the brain has adequate exposure to the anti-amyloid therapeutic antibodies to achieve any potential therapeutic efficacy. It is therefore hard to determine, whether simultaneously administration of another pharmacological agent that improves BBB integrity in such context may be beneficial by reducing brain tissue exposure to neurotoxic and proinflammatory substances, or would hamper therapeutic antibodies from exerting the desirable effects. We have now further discussed these considerations in the revised manuscript (page 13, paragraph 3).

We also agree with the reviewer that we have to consider mechanisms beyond BBB protection to explain the beneficial effects of GLP-1R agonist found in AD. Specifically, our data support that the amelioration of age-related microglial priming plays a role (Supplementary Fig. 9). This may be

an underlying mechanism accounting for the BBB integrity rescue we found, and/or as a consequence of EC transcriptomic reversal and BBB improvement with GLP-1R agonist treatment (also see response to comment 5).

4. In Fig 4, the authors identified similar expression changes in normal aged mouse and human AD brains. The changes of some EC-enriched genes in aged mouse brain, had similar direction with AD human brain data, but not the same with aged human brain data, e.g. *Slc2a1*. But aging is not the same as AD. What is the possible meaning of this? Does this mean the normal aged mouse can model human AD? Are there any genes show concordant changes in aged mouse and aged human brain, but different in AD human brain? These might be more related with AD pathology.

We thank the reviewer for the comments, and we agree that (1) the significance of concordant expression changes in aged mouse and human AD brain should be discussed further, and (2) genes whose expression changes are concordant in aged brains across species but discordant in human AD are worth looking for.

For genes that exhibit similar changes for aged mouse (compared to young adult mouse) and human AD brains (compared to age-matched controls) and have significant functional implications at the neurovasculature, we indeed propose that they may imply similar functional changes at the endothelium in aged mice and human AD brains. Currently, the field base pathophysiological study and therapeutic testing largely on experimenting in familial AD mouse models, which are unable to model all aspects of human AD pathology. Our findings therefore carry the implication that aged mouse brains may serve well on modelling specific aspects of neurovascular dysfunction found in human AD. We have strengthened our discussions on this point (page 12, paragraph 1).

We agree with the reviewer that genes with concordant expression change in aged mouse and human brains and yet exhibit discordance with human AD brains may reflect important changes that diverge in AD from normal ageing. This point is now acknowledged in the revised manuscript (page 8, paragraph 2). With our current dataset, we noted only one such gene (*Epas1*, an EC and pericyte-enriched gene). We also wish to point out however that *EPAS1* was found to be elevated in human AD entorhinal cortex ECs in a recent study (Grubman et al., (2019)) and yet downregulated in the Allen Brain human AD dataset that we analyzed, and the reason(s) of such inconsistency is not fully clear. We believe that further single-cell transcriptomic profiling with adequate vascular cell sampling will be required to identify and verify the expression changes of more such genes. More studies on humans will then help better design AD mouse model on top of aged mice.

5. The rather sudden “jump” to exenatide is unclear. Not much rationale is provided for GLP-1R. Why choose this target? It is mentioned that GLP-1R agonists have been reported to provide neuronal protection, but is there also some justification for selecting this for BBB and endothelium? In fact, GLP-1R may be a universally expressed gene. Why this focus in aged mouse brain, or aged or AD human brains?

We apologize for our explanations on the rationale of testing the efficacy of an GLP-1R agonist were inadequate. In the revised manuscript, we have now explained in more detail the heuristics behind our choice . In short, we made the following observations: (1) human clinical trials have found potential efficacies of GLP-1R agonists in two neurodegenerative conditions, improving brain metabolism in AD and delaying motor deterioration in Parkinson’s disease (PD) (refs 39 and 40: Gejl et al. (2016), Athauda et al. (2017)), (2) in animal experimentation, GLP-1R agonists were

found to be efficacious in ameliorating the progression of neuropathology in animal models of AD, PD, amyotrophic lateral sclerosis and stroke (refs 33-38: Kim et al. (2009), Li et al. (2009, 2010, 2012), McClean et al. (2011), Yun et al. (2018)), (3) the use of GLP-1R agonists in diabetic patients reduce the incidence of cardiovascular events (e.g. myocardial infarction, stroke) (reviewed in ref 41: Kristensen et al. (2019)), while animal tests revealed vascular protective effects in peripheral organs (ref 42 and 43: Noyan-Ashraf et al. (2009), Helmstadter et al. (2019)), and (4) we observed that impaired energy metabolism and altered immune/cytokine signalling were implicated especially in capillary ECs by the scRNA-seq data, which may express GLP-1R (see below). At other body sites (e.g. the myocardium), GLP-1R signalling has been reported to alter energy metabolism and exert immunomodulatory effects (refs 44-48: Arakawa et al. (2010), Yusta et al. (2015), Bruen et al. (2017), Vinue et al. (2017), Filippidou et al. (2020)). With these considerations, we thus postulated that GLP-1R agonist may have neurovascular protective effects that could help explain its general applicability as potential therapeutics for multiple neurodegenerative conditions with age-associated neurovascular defects as a shared pathological component. These points are now discussed in the Introduction (page 3, paragraph 1) and Results sections (page 9, paragraph 1).

The expression pattern of GLP-1R in the brain is still an active area of research, whereby past literature has revealed its presence in subsets of neurons, microglia, while peripherally it may be expressed in vascular cells (refs 37, 49-51: Yun et al. (2018), Cork et al. (2015), Lovshin et al. (2015), Jensen et al. (2018)). We have examined the expression pattern of GLP-1R in the mouse brain by RNA FISH ourselves, apart from referencing previous literature (page 3, paragraph 1). Note that in both our scRNA-seq dataset and that by Vanlandewijck, et al. (2018), we did not find GLP-1R transcripts in ECs. RNA FISH revealed almost no localization of GLP-1R transcripts to ECs (see attached Rebuttal Fig. 1 at the end of this response letter). It thus appeared that ECs had low expression of GLP-1R, if any. We have discussed the potential mechanisms that may underlie the transcriptomic reversal and BBB protective effects of GLP-1R agonist we found (page 13, paragraph 2). Indeed, there are multiple possible underlying mechanisms of action, whereby we postulate that indirect actions via microglia or modulation of compositions in the peripheral circulation are equally important to consider. This is also supported by our observation that the upregulation of microglial ageing- and neurodegenerative disease-associated transcripts was decreased by GLP-1R agonist treatment (Supplementary Fig. 9, also see response to comment 1). We wish to hold on to this and publish separately when we have convincingly identified the exact mechanism(s).

6. The cell number of endothelial cells from 5 aged mice appear to be lower than the one from 5 young mice. It seems unclear whether this was due to a batch effect, or just because aged brains have a fewer number of endothelial cells. Vascular density may be important when calculating dextran data? When the authors using dextran to detect permeability of BBB, did they check the density of the vessels? It seems the vessels density might be higher in the aged mouse's cortex than the young mouse?

Thanks to the reviewer for raising points regarding (1) the difference in number of cells obtained from aged versus young mouse brains and (2) potential impact of differences in vascular length density on the BBB assay results.

We adopted identical brain extraction, tissue digestion and cell dissociation protocols to the young and aged mouse groups. For each batch of sequencing experiments (three in total), a paired design was adopted, whereby both age groups were present. The difference in cell number obtained

was therefore not due to a batch effect. Instead, this may reflect a potential decrease in overall vascular density with age (ref 91: Riddle et al. (2003)), differences in brain tissue reactions to the same digestion and dissociation protocol, and variability in cell counting procedure (due to the presence of more myelin debris in aged group) before loading cell suspensions onto the single-cell encapsulation system, hence impacting the number of cells sampled. The EC numbers from each batch are now mentioned in the revised manuscript for readers' reference (page 16, paragraph 3).

As suggested by the reviewer, we have also quantified cortical vascular length density from the image stacks obtained from the three experimental groups (young adult, aged and exenatide-treated aged mouse groups). We however did not find differences across the different groups in the region (i.e. the cortex) where we assayed BBB leakage (Fig. 4c; page 9, paragraph 2), we thus believe this would not confound our analysis and comparison.

7. The authors may need to clarify whether they prepared single-cell RNAseq libraries from mice that were also used for two-photon imaging studies. The surgery for cranial window (and the procedures of two-photon imaging) may change the mRNA expression patterns in brain cells.

We used different mice for scRNA-seq and *in vivo* imaging experiments, therefore none of the mice that underwent cranial window implantation or imaging was used for scRNA-seq library preparation. We have now clarified this point in the methods session (page 15, paragraph 1).

8. In Fig 5b: they stated “n=9 image stacks from 3 mice for each group”. Does this mean that the values for mean +/- SEM in Fig 5b were from n=9? If yes, probably, they need to recalculate the average (and SEM) from 3 mice (the FC value for each mouse should be from the average FC value of 3 image stacks in each mouse?).

We have redone the analysis based on the reviewer's advice (Fig. 4b).

9. The authors should give a confirmation on cell purity: high gene expression values for genes with EC-specific expression and very low or undetectable expression for other cell- type-specific markers of the central nervous system.

We thank the reviewer for this suggestion. In our original version of the manuscript, data presented in Supplementary Fig. 1a indeed showed that the EC cluster identified had high expression of *Cldn5* and did not express numerous other cell-type-specific marker genes (including *Kcnj8*, *Acta2*, *Ctss*, *Ntsr2*, *Pdgfra*, *Cldn11* etc., see Supplementary Fig. 1a for details). To make this point more explicit and better present the data, we now refer to this point more clearly in the main text (page 3, paragraph 3). We have also revised Supplementary Fig. 1a to show that the ECs had high expression of several other EC-specific marker genes including *Flt1*, *Pecam1* and *Cdh5*, in addition to *Cldn5* (Supplementary Fig. 1a). These results thus concluded our claim on the purity of the ECs.

10. Some key signals should be confirmed with protein.

Thanks for the advice. We have now carried out validation experiments to confirm the differential expression of selected genes involved in key pathways or regulation of neurovascular functions (e.g. adherens junction, VEGF and TGF- β signalling-related genes, key transporter genes) in pooled endothelial cells (ECs) isolated from mouse brains by immunopanning (Fig. 2c, also see Methods). We verified the differential expression of several aged brain EC-upregulated (*Flt1*, *Klf6*,

Smad7) and downregulated (Mfsd2a) genes (Fig. 2d) by qPCR, and for protein targets for which we identified suitable antibodies (LEF1, SMAD7 and MFSD2A), we also validated the expression changes at the protein level by western blot (Fig. 2e). These results are now all incorporated in the results session (page 6, paragraph 3).

11. & 12. The authors used whole brain to prepare the libraries, but did not discuss the brain-region-related heterogeneity of the data. Especially because aging affects specific brain areas differently, the authors might wish to address whether region-specific (such as cortices and hippocampi etc) display key differences in gene expression patterns upon aging. If space allows, it may be nice to provide some discussion into this point. It may be nice to discuss potential sex differences because this study used only male mice.

We fully agree with the reviewer on these comments and we apologize that our original discussions on this point towards the end of the Discussion section was not adequate. We have now extended and further elaborated on this point, that the regional heterogeneity of cellular subtypes and differential expressions in ageing may require further experimentation to uncover and could have important pathophysiological implications in relation to diseases with regional vulnerability (e.g. the frontal and temporal cortices in frontotemporal lobar degeneration) (page 14, paragraph 1). On the other hand, we also thank the reviewer for reminding us on the importance of sex in relation to ageing and neurodegeneration. Indeed, it is known that aged females have different neurovascular and neurodegenerative disease risk profiles, and that brain transcriptomic alterations also exhibits sex-specificity in AD (e.g. as reported in a recent study, ref 117: Mathys et al. (2019)). We have also discussed this point in the revised manuscript (page 14, paragraph 1).

13. Finally, how “specific” are the endothelial patterns found in this study? There are many other papers that have previously mapped aging transcriptomes for brain endothelium, astrocytes, microglia etc (e.g. Guo et al, Neurobiol Dis 2019; Boisvert et al, Cell Rep 2018; Olah et al, Nat Comm 2018). It is a bit surprising that these previous papers were not referenced or discussed.

We thank the reviewer for reminding us to discuss our findings in relation to these previous studies. We have now strengthened our discussions on the relationship of our findings to studies on ageing transcriptomes by Boisvert et al. (2018) on astrocytes, Olah et al. (2018) on microglia, Guo et al. (2019) on ECs (page 11, paragraph 2 and 3), Dulken et al. (2019) on subventricular zone neurogenic niche and Yousef et al. (2019) on hippocampal ECs (page 11, paragraph 2). Indeed, this allows us to better relate to other studies findings, and discuss both shared/related and specific functional changes implicated in aged brain ECs found.

In brief, Boisvert et al. (2018) found that aged astrocyte transcriptomes from multiple brain regions partially exhibit reactive phenotypes with upregulation of complement and MHC-related genes (page 11, paragraph 2). Dulken et al. (2019) reported that multiple cell types in the SVZ including ECs exhibit interferon- γ -signalling related genes upregulation that mediates T cell infiltration (discussed on page 11, paragraph 2). A closely related study by Yousef et al. (2019) also identified immune-related gene differential expression in aged brain hippocampal ECs, whereby Vcam1 upregulation accounts for increased immune cell tethering and endothelial activation at the brain vasculature (referenced to and discussed on page 2, paragraph 3; page 6, paragraph 1; page 11, paragraph 2). From our dataset, apart from immune cell transmigration pathway enrichment in the upregulated DEGs of capillary ECs, we mainly found interleukin pathway enrichment in multiple

EC subtypes. These EC changes may interact with the changes in other cell types in the aged brain identified by Boisvert et al. (2018), Dulken et al. (2019) and Yousef et al. (2019) and jointly mediate BBB breakdown, neuroinflammation and neurodegenerative disease susceptibility (discussed on page 11, paragraph 2). On the other hand, Guo et al. (2018) found that by GSEA analysis, immune suppression of ECs is implicated in aged brain ECs. Integrating findings from our study and these studies, we speculate that in the aged brain, normal physiological immune responses of brain ECs may be hampered as maladaptive immune responses prevails. These are all now discussed in the revised manuscript (page 11, paragraph 2).

Finally, Olah et al. (2018) reported an overrepresentation of AD GWAS genes among aged brain microglia, while we found overrepresentation in aged brain capillary ECs. Comparing our list of AD GWAS genes among the DEGs of aged brain ECs (Fig. 3a) against that of Olah et al. (2018), we noted that the gene lists are non-overlapping. This may imply that the subset that we have identified have a potential vascular linkage to AD, while that by Olah et al. (2018) have stronger microglia-related influence on AD development. This is also discussed in the revised manuscript (page 11, paragraph 3).

Rebuttal Fig. 1 RNA fluorescent *in situ* hybridization of Glp1r transcripts in young adult (3 months old, upper two rows) and aged (18 months old, lower two rows) mouse brain sections, with lectin and DAPI co-staining for visualization of endothelium and cell nuclei.

Reviewers' Comments:

Reviewer #1:

Remarks to the Author:

Zhao et al. have provided a thorough and thoughtful reply to the comments of both reviewers. I appreciate the considerable extra effort to validate changes to brain endothelial cells with other methods, and the qPCR, western blot, and FISH provide considerable evidence to back up changes observed by scRNAseq, particularly during this time. The flow of the article is also much improved with the additional acknowledgement of recent publications by the Wyss-Coray lab, and this work provides an excellent complement to their work detailing changes to brain endothelium during aging. Finally, the additional analysis of microglia activation, and discussion around mechanistic aspects of GLP1R treatment provide greater context. This is timely and exciting work that provides exquisite details of changes to the neurovasculature during aging, as well as potential therapeutic avenues, and I recommend its acceptance.

Reviewer #2:

Remarks to the Author:

To Authors: This is a study with lots of data. Much of the data may replicate what has been published in the literature in terms of aging vascular signatures (e.g. Boisvert, Cell Rep 2018; Olah, Nat Comm 2018; Guo Neurobiol Dis 2019; Tabula Muris Consortium, BioRxiv 2019). There are some remaining concerns.

1. The authors still appear to conclude the major response in the aging vascular system is centered on the BBB, and that exenatide efficacy is mediated by BBB rescue. One may still worry that the data might not fully support these two major conclusions. Many other non-BBB pathways are affected. Exenatide surely affects non-BBB/non-endothelial targets in vivo.
2. There may still be several inconsistent findings. Many BBB genes move in opposite directions in aged mouse versus aged humans. Overall, the aged normal mouse is more similar to the aged AD human but not the normal aged human. Why?
3. Finally, a minor technical question please. The authors replotted Fig 5b as the new Fig 4b. The original Fig 5b stated that "n=9 image stacks from 3 mice for each group". If it is true that this meant 3 image stacks from each mouse, and 3 mice total, then it was pointed out that calculating means and SEM from n=9 data points was incorrect. What should be done is to first calculate a mean of 3 image stacks per mouse, and then calculate the group average with only n=3 data points. If so, shouldn't the SEM (SD divided by square root of n) be very different for n=9 versus n=3? It is surprising that the replotted means and SEM are almost exactly the same.

Reviewer #1 (Remarks to the Author):

Zhao et al. have provided a thorough and thoughtful reply to the comments of both reviewers. I appreciate the considerable extra effort to validate changes to brain endothelial cells with other methods, and the qPCR, western blot, and FISH provide considerable evidence to back up changes observed by scRNAseq, particularly during this time. The flow of the article is also much improved with the additional acknowledgement of recent publications by the Wyss-Coray lab, and this work provides an excellent complement to their work detailing changes to brain endothelium during aging. Finally, the additional analysis of microglia activation, and discussion around mechanistic aspects of GLP1R treatment provide greater context. This is timely and exciting work that provides exquisite details of changes to the neurovasculature during aging, as well as potential therapeutic avenues, and I recommend its acceptance.

We thank the reviewer for the encouraging remarks and providing critical comments that helped us in substantially improving the manuscript. We are delighted that the reviewer recommended acceptance of the manuscript.

Reviewer #2 (Remarks to the Author):

To Authors: This is a study with lots of data. Much of the data may replicate what has been published in the literature in terms of aging vascular signatures (e.g. Boisvert, Cell Rep 2018; Olah, Nat Comm 2018; Guo Neurobiol Dis 2019; Tabula Muris Consortium, BioRxiv 2019). There are some remaining concerns.

We thank the reviewer for acknowledging our efforts, and for making important suggestions that helped us in significantly improving the manuscript.

1. The authors still appear to conclude the major response in the aging vascular system is centered on the BBB, and that exenatide efficacy is mediated by BBB rescue. One may still worry that the data might not fully support these two major conclusions. Many other non-BBB pathways are affected. Exenatide surely affects non-BBB/non-endothelial targets in vivo.

We agree with the reviewer that altered pathways other than those that impact the blood-brain barrier (BBB) are implicated in the ageing-associated transcriptomic changes we found, and that cell types other than endothelial cells (ECs) are also affected by GLP-1R agonist treatment. For this reason, we thoroughly revised our manuscript in the last revision, to (i) more accurately reflect the diversity of functional roles of the enriched pathways among aged brain EC differential expressions (by revising Fig. 2a annotation to “*Signalling pathways with diverse functions including vascular and BBB regulation*” and the main text at appropriate places as enlisted in the previous rebuttal); (ii) adjust our discussions on the potential roles of altered immune-related pathways in the aged brain endothelium, as well as their relationship to other neurovascular cell types; (iii) supplement with extra experiments and analyses assaying the effects of exenatide treatment on microglia (Supplementary Fig. 9). In agreement with the reviewer’s comment that “non-BBB/non-endothelial targets” are also affected, we found GLP-1R agonist treatment attenuated microglial expression of ageing- and neurodegenerative disease-associated transcripts. This could be an additional mechanism underlying the BBB-protective effect found, and the efficacies of GLP-1R agonists reported in human Alzheimer’s disease (AD) and Parkinson’s disease (PD) trials (refs 39-40: Gejl, M. et al., Frontiers in aging neuroscience (2016); Athauda, D. et al., Lancet (2017)).

Therefore, while we discussed on BBB function, which is solidly established to be important in age-related neurodegenerative diseases (reviewed in e.g. Strooper et al., Cell (2016); Iadecola, Neuron (2017); Sweeney et al., Nature Neuroscience (2018), Alzheimer’s & Dementia (2019)), we also acknowledge the importance of pathways other than BBB regulatory ones and non-vascular cell types.

To address the remaining concern, we have now further extended our discussions on this point (page 13, paragraph 2): *“As GLP-1R is broadly expressed in diverse cell types, and GLP-1R agonists can cross the BBB, the observed neurovascular benefits may depend on diverse mechanisms, such as transcriptomic reversal in aged brain ECs, reduced microglial activation, modulation of peripheral immune cells and altered compositions in the circulation. Indeed, our finding that GLP-1R agonist treatment reduces the expression of ageing- and neurodegenerative disease-associated transcripts in microglia, which has been reported to express GLP-1R, suggests that modulation of microglia-dependent neuroinflammatory pathways is an important effect of GLP-1R agonists. Certainly, both neurovascular and neuroimmune mechanisms could be underlying the efficacies of GLP-1R agonists reported in human AD and PD trials.”*

2. There may still be several inconsistent findings. Many BBB genes move in opposite directions in aged mouse versus aged humans. Overall, the aged normal mouse is more similar to the aged AD human but not the normal aged human. Why?

Thanks for raising this discussion point. It is indeed very intriguing that the aged mouse brain EC expression changes partially resemble that of the human AD brain, but exhibit significant discordance with the normal aged human brain. We speculate that there are multiple possible explanations for these observations, including: (i) inter-species differences, whereby the aged mouse brain vasculature is more prone to expression and functional changes that predispose the nervous system to degenerative changes, and (ii) the potential bias from profiling non-neuronal cells' expression changes (raised by reviewer 1 in the last revision) in the human brain by bulk RNA-seq, which could lead to the exaggerated upregulation and masked downregulation of EC-enriched genes, accounting partly for the discordance. This was the reason that we moved the data presented in Fig. 4a-b of the initial version to supplementary information (Supplementary Fig. 6), and acknowledged the potential caveat in the manuscript (see page 8, paragraph 2).

In the revised manuscript, further discussions concerning this point have been added (page 12, paragraph 2): *“The mechanism(s) for the partial resemblance of aged mouse brain vascular changes to that in the human AD brain and some discordances with the normal aged human brain are yet to be determined. These could be reflecting inter-species differences, whereby the aged mouse brain vasculature is more prone to expression and functional alterations that predispose the nervous system to degenerative changes.”*

3. Finally, a minor technical question please. The authors replotted Fig 5b as the new Fig 4b. The original Fig 5b stated that “n=9 image stacks from 3 mice for each group”. If it is true that this meant 3 image stacks from each mouse, and 3 mice total, then it was pointed out that calculating means and SEM from n=9 data points was incorrect. What should be done is to first calculate a mean of 3 image stacks per mouse, and then calculate the group average with only n=3 data points. If so, shouldn't the SEM (SD divided by square root of n) be very different for n=9 versus n=3? It is surprising that the replotted means and SEM are almost exactly the same.

Thanks for the technical comment, which prompted us to clarify our presentation further. In the previous revision, we adopted the suggestion on averaging for each animal first before calculating the mean and SEM for each group. We have included all the numbers underlying Fig. 4b in the Source Data file, which also shows the calculations to obtain, for each group, the mean and SEM of (i) values from all 9 image stacks, and (ii) means of individual animals. As we had the same number of image stacks (i.e. 3) taken for each animal, the means obtained by either way of calculation must be exactly the same. For SEM, since averaging for each animal reduces the scatter of the numbers

(i.e. for each group, the means of the 3 individual animals are less scattered than the values from the 9 image stacks), hence dividing the SD of the 3 animals' means by $\sqrt{3}$ resulted in similar values as the SD of the values from the 9 image stacks divided by $\sqrt{9}$. These numbers are shown in the Rebuttal Table 1 below (see numbers highlighted in red and underlined) and can be verified by checking the corresponding numbers and formulae in the Source Data excel file. To ensure clarity of the sample size description, we have also revised the main text (page 9, paragraph 2) and Fig. 4b legends to “3 image stacks were acquired to obtain the mean for each animal, $n = 3$ mice for each group”.

Rebuttal Table 1. Source data underlying Fig. 4b (also in the Source Data file).

Fig. 4b	Animal	Stack	Young	Aged	Aged (Exenatide-treated)
Fold change of extravasated dextran (40 kDa) relative to young adult group mean	1	1	0.794	12.823	7.643
		2	0.691	18.482	7.588
		3	1.077	20.383	7.573
	2	1	0.796	13.816	7.086
		2	1.039	12.002	6.262
		3	1.239	13.794	7.865
	3	1	1.247	17.172	7.289
		2	1.171	18.238	8.636
		3	0.946	15.231	7.379
Stats	Mean	1.000	15.771	7.480	
	SD	0.205	2.907	0.634	
	SEM	0.068	0.969	0.211	
Stats (calculating the mean of each animal first)	Avg1	0.854	17.229	7.601	
	Avg2	1.025	13.204	7.071	
	Avg3	1.121	16.881	7.768	
	Mean	1.000	15.771	7.480	
	SD	0.135	2.230	0.364	
	SEM	0.078	1.288	0.210	

Reviewers' Comments:

Reviewer #2:

Remarks to the Author:

No further suggestions. However, there is still a major worry that there is still no way to interpret the lack of consistent "directionality" of BBB-related genes in the mouse versus human data sets and the non-BBB actions of exenatide. Ultimately, there is a lot of data. But it remains unclear whether this study can really show a clear conclusion about the use of exenatide to treat AD via a BBB mechanism.

Reviewer #2 (Remarks to the Author):

No further suggestions. However, there is still a major worry that there is still no way to interpret the lack of consistent "directionality" of BBB-related genes in the mouse versus human data sets and the non-BBB actions of exenatide. Ultimately, there is a lot of data. But it remains unclear whether this study can really show a clear conclusion about the use of exenatide to treat AD via a BBB mechanism.

We wish to express our gratitude to the reviewer again for having made important suggestions that helped us substantially improved the manuscript.

As discussed during previous rounds of revision, we acknowledge that while some age-related changes we found in the aged mouse brain do appear to be more concordant with the human AD brain, it likely only partially resemble that of both the human AD and normal aged brain. We agree that this may reflect inter-species differences and require further studies of human vascular cell transcriptomic profiling across age to address.

On the other hand, we do believe that our data provide strong evidence for the protective effects of GLP-1R agonists against age-related transcriptomic and functional changes at the blood-brain barrier. This study will therefore bring insights to the field on further mechanistic studies, as well as potential clinical trials testing the efficacy of GLP-1R agonists in the treatment of age-related vasculopathy – a contributing factor to AD.